# Towards deep learning with segregated dendrites

Jordan Guerguiev[1,2], Timothy P Lillicrap[3], Blake A Richards[1,2,4]*

[1]Department of Biological Sciences, University of Toronto Scarborough, Toronto, Canada; [2]Department of Cell and Systems Biology, University of Toronto, Toronto, Canada; [3]DeepMind, London, United Kingdom; [4]Learning in Machines and Brains Program, Canadian Institute for Advanced Research, Toronto, Canada

**Abstract** Deep learning has led to significant advances in artificial intelligence, in part, by adopting strategies motivated by neurophysiology. However, it is unclear whether deep learning could occur in the real brain. Here, we show that a deep learning algorithm that utilizes multi-compartment neurons might help us to understand how the neocortex optimizes cost functions. Like neocortical pyramidal neurons, neurons in our model receive sensory information and higher-order feedback in electrotonically segregated compartments. Thanks to this segregation, neurons in different layers of the network can coordinate synaptic weight updates. As a result, the network learns to categorize images better than a single layer network. Furthermore, we show that our algorithm takes advantage of multilayer architectures to identify useful higher-order representations—the hallmark of deep learning. This work demonstrates that deep learning can be achieved using segregated dendritic compartments, which may help to explain the morphology of neocortical pyramidal neurons.

DOI: https://doi.org/10.7554/eLife.22901.001

## Introduction

Deep learning refers to an approach in artificial intelligence (AI) that utilizes neural networks with multiple layers of processing units. Importantly, deep learning algorithms are designed to take advantage of these multi-layer network architectures in order to generate hierarchical representations wherein each successive layer identifies increasingly abstract, relevant variables for a given task (*Bengio and LeCun, 2007*; *LeCun et al., 2015*). In recent years, deep learning has revolutionized machine learning, opening the door to AI applications that can rival human capabilities in pattern recognition and control (*Mnih et al., 2015*; *Silver et al., 2016*; *He et al., 2015*). Interestingly, the representations that deep learning generates resemble those observed in the neocortex (*Kubilius et al., 2016*; *Khaligh-Razavi and Kriegeskorte, 2014*; *Cadieu et al., 2014*), suggesting that something akin to deep learning is occurring in the mammalian brain (*Yamins and DiCarlo, 2016*; *Marblestone et al., 2016*).

Yet, a large gap exists between deep learning in AI and our current understanding of learning and memory in neuroscience. In particular, unlike deep learning researchers, neuroscientists do not yet have a solution to the 'credit assignment problem' (*Rumelhart et al., 1986*; *Lillicrap et al., 2016*; *Bengio et al., 2015*). Learning to optimize some behavioral or cognitive function requires a method for assigning 'credit' (or 'blame') to neurons for their contribution to the final behavioral output (*LeCun et al., 2015*; *Bengio et al., 2015*). The credit assignment problem refers to the fact that assigning credit in multi-layer networks is difficult, since the behavioral impact of neurons in early layers of a network depends on the downstream synaptic connections. For example, consider the behavioral effects of synaptic changes, that is long-term potentiation/depression (LTP/LTD), occurring between different sensory circuits of the brain. Exactly how these synaptic changes will impact

**\*For correspondence:**
blake.richards@utoronto.ca

**Competing interests:** The authors declare that no competing interests exist.

**eLife digest** Artificial intelligence has made major progress in recent years thanks to a technique known as deep learning, which works by mimicking the human brain. When computers employ deep learning, they learn by using networks made up of many layers of simulated neurons. Deep learning has opened the door to computers with human – or even super-human – levels of skill in recognizing images, processing speech and controlling vehicles. But many neuroscientists are skeptical about whether the brain itself performs deep learning.

The patterns of activity that occur in computer networks during deep learning resemble those seen in human brains. But some features of deep learning seem incompatible with how the brain works. Moreover, neurons in artificial networks are much simpler than our own neurons. For instance, in the region of the brain responsible for thinking and planning, most neurons have complex tree-like shapes. Each cell has 'roots' deep inside the brain and 'branches' close to the surface. By contrast, simulated neurons have a uniform structure.

To find out whether networks made up of more realistic simulated neurons could be used to make deep learning more biologically realistic, Guerguiev et al. designed artificial neurons with two compartments, similar to the 'roots' and 'branches'. The network learned to recognize hand-written digits more easily when it had many layers than when it had only a few. This shows that artificial neurons more like those in the brain can enable deep learning. It even suggests that our own neurons may have evolved their shape to support this process.

If confirmed, the link between neuronal shape and deep learning could help us develop better brain-computer interfaces. These allow people to use their brain activity to control devices such as artificial limbs. Despite advances in computing, we are still superior to computers when it comes to learning. Understanding how our own brains show deep learning could thus help us develop better, more human-like artificial intelligence in the future.

DOI: https://doi.org/10.7554/eLife.22901.002

behavior and cognition depends on the downstream connections between the sensory circuits and motor or associative circuits (*Figure 1A*). If a learning algorithm can solve the credit assignment

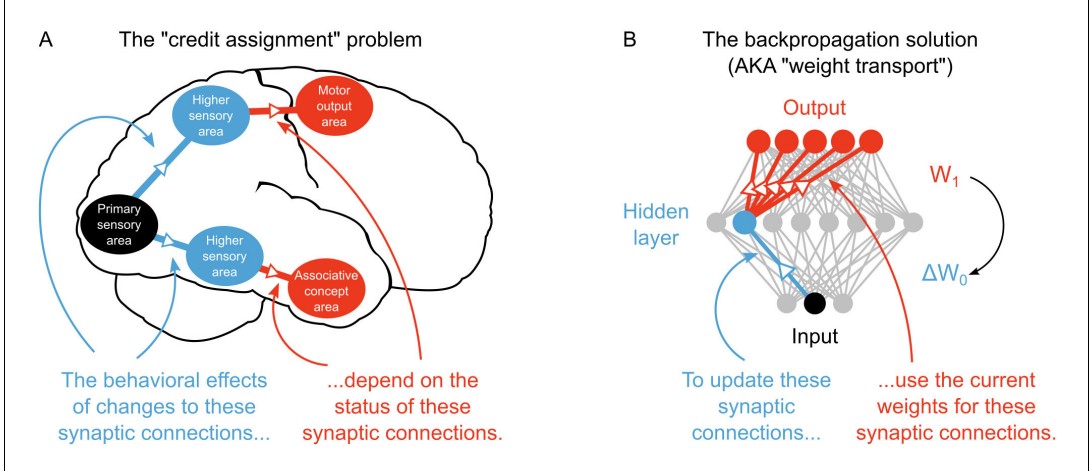

**Figure 1.** The credit assignment problem in multi-layer neural networks. (**A**) Illustration of the credit assignment problem. In order to take full advantage of the multi-circuit architecture of the neocortex when learning, synapses in earlier processing stages (blue connections) must somehow receive 'credit' for their impact on behavior or cognition. However, the credit due to any given synapse early in a processing pathway depends on the downstream synaptic connections that link the early pathway to later computations (red connections). (**B**) Illustration of weight transport in backpropagation. To solve the credit assignment problem, the backpropagation of error algorithm explicitly calculates the credit due to each synapse in the hidden layer by using the downstream synaptic weights when calculating the hidden layer weight changes. This solution works well in AI applications, but is unlikely to occur in the real brain.

DOI: https://doi.org/10.7554/eLife.22901.003

problem then it can take advantage of multi-layer architectures to develop complex behaviors that are applicable to real-world problems (*Bengio and LeCun, 2007*). Despite its importance for real-world learning, the credit assignment problem, at the synaptic level, has received little attention in neuroscience.

The lack of attention to credit assignment in neuroscience is, arguably, a function of the history of biological studies of synaptic plasticity. Due to the well-established dependence of LTP and LTD on presynaptic and postsynaptic activity, current theories of learning in neuroscience tend to emphasize Hebbian learning algorithms (*Dan and Poo, 2004*; *Martin et al., 2000*), that is, learning algorithms where synaptic changes depend solely on presynaptic and postsynaptic activity. Hebbian learning models can produce representations that resemble the representations in the real brain (*Zylberberg et al., 2011*; *Leibo et al., 2017*) and they are backed up by decades of experimental findings (*Malenka and Bear, 2004*; *Dan and Poo, 2004*; *Martin et al., 2000*). But, current Hebbian learning algorithms do not solve the credit assignment problem, nor do global neuromodulatory signals used in reinforcement learning (*Lillicrap et al., 2016*). As a result, deep learning algorithms from AI that can perform multi-layer credit assignment outperform existing Hebbian models of sensory learning on a variety of tasks (*Yamins and DiCarlo, 2016*; *Khaligh-Razavi and Kriegeskorte, 2014*). This suggests that a critical, missing component in our current models of the neurobiology of learning and memory is an explanation of how the brain solves the credit assignment problem.

However, the most common solution to the credit assignment problem in AI is to use the back-propagation of error algorithm (*Rumelhart et al., 1986*). Backpropagation assigns credit by *explicitly* using current downstream synaptic connections to calculate synaptic weight updates in earlier layers, commonly termed 'hidden layers' (*LeCun et al., 2015*) (*Figure 1B*). This technique, which is sometimes referred to as 'weight transport', involves non-local transmission of synaptic weight information between layers of the network (*Lillicrap et al., 2016*; *Grossberg, 1987*). Weight transport is clearly unrealistic from a biological perspective (*Bengio et al., 2015*; *Crick, 1989*). It would require early sensory processing areas (e.g. V1, V2, V4) to have precise information about *billions* of synaptic connections in downstream circuits (MT, IT, M2, EC, etc.). According to our current understanding, there is no physiological mechanism that could communicate this information in the brain. Some deep learning algorithms utilize purely Hebbian rules (*Scellier and Bengio, 2016*; *Hinton et al., 2006*). But, they depend on feedback synapses that are symmetric to feedforward synapses (*Scellier and Bengio, 2016*; *Hinton et al., 2006*), which is essentially a version of weight transport. Altogether, these artificial aspects of current deep learning solutions to credit assignment have rendered many scientists skeptical of the proposal that deep learning occurs in the real brain (*Crick, 1989*; *Grossberg, 1987*; *Harris, 2008*; *Urbanczik and Senn, 2009*).

Recent findings have shown that these problems may be surmountable, though. *Lillicrap et al. (2016)*, *Lee et al., 2015* and *Liao et al., 2015* have demonstrated that it is possible to solve the credit assignment problem even while avoiding weight transport or symmetric feedback weights. The key to these learning algorithms is the use of feedback signals that convey enough information about credit to calculate local error signals in hidden layers (*Lee et al., 2015*; *Lillicrap et al., 2016*; *Liao et al., 2015*). With this approach it is possible to take advantage of multi-layer architectures, leading to performance that rivals backpropagation (*Lee et al., 2015*; *Lillicrap et al., 2016*; *Liao et al., 2015*). Hence, this work has provided a significant breakthrough in our understanding of how the real brain might do credit assignment.

Nonetheless, the models of *Lillicrap et al. (2016)*, *Lee et al., 2015* and *Liao et al., 2015* involve some problematic assumptions. Specifically, although it is not directly stated in all of the papers, there is an implicit assumption that there is a separate feedback pathway for transmitting the information that determines the local error signals (*Figure 2A*). Such a pathway is required in these models because the error signal in the hidden layers depends on the difference between feedback that is generated in response to a purely feedforward propagation of sensory information, and feedback that is guided by a teaching signal (*Lillicrap et al., 2016*; *Lee et al., 2015*; *Liao et al., 2015*). In order to calculate this difference, sensory information must be transmitted *separately* from the feedback signals that are used to drive learning. In single compartment neurons, keeping feedforward sensory information separate from feedback signals is impossible without a separate pathway. At face value, such a pathway is possible. But, closer inspection uncovers a couple of difficulties with such a proposal.

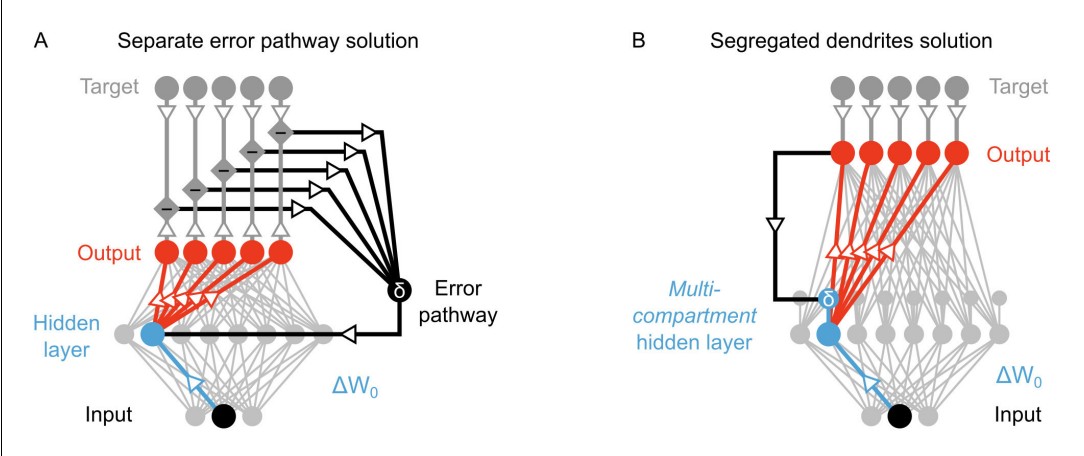

**Figure 2.** Potential solutions to credit assignment using top-down feedback. (**A**) Illustration of the implicit feedback pathway used in previous models of deep learning. In order to assign credit, feedforward information must be integrated separately from any feedback signals used to calculate error for synaptic updates (the error is indicated here with $\delta$). (**B**) Illustration of the segregated dendrites proposal. Rather than using a separate pathway to calculate error based on feedback, segregated dendritic compartments could receive feedback and calculate the error signals locally.

DOI: https://doi.org/10.7554/eLife.22901.004

First, the error signals that solve the credit assignment problem are not global error signals (like neuromodulatory signals used in reinforcement learning). Rather, they are *cell-by-cell* error signals. This would mean that the feedback pathway would require some degree of pairing, wherein each neuron in the hidden layer is paired with a feedback neuron (or circuit). That is not impossible, but there is no evidence to date of such an architecture in the neocortex. Second, the error signal in the hidden layer is signed (i.e. it can be positive or negative), and the sign determines whether LTP or LTD occur in the hidden layer neurons (*Lee et al., 2015*; *Lillicrap et al., 2016*; *Liao et al., 2015*). Communicating signed signals with a spiking neuron can theoretically be done by using a baseline firing rate that the neuron can go above (for positive signals) or below (for negative signals). But, in practice, such systems are difficult to operate with a single neuron, because as the error gets closer to zero any noise in the spiking of the neuron can switch the sign of the signal, which switches LTP to LTD, or *vice versa*. This means that as learning progresses the neuron's ability to communicate error signs gets *worse*. It would be possible to overcome this by using many neurons to communicate an error signal, but this would then require many error neurons for *each* hidden layer neuron, which would lead to a very inefficient means of communicating errors. Therefore, the real brain's specific solution to the credit assignment problem is unlikely to involve a separate feedback pathway for cell-by-cell, signed signals to instruct plasticity.

However, segregating the integration of feedforward and feedback signals does not require a separate pathway if neurons have more complicated morphologies than the point neurons typically used in artificial neural networks. Taking inspiration from biology, we note that real neurons are much more complex than single-compartments, and different signals can be integrated at distinct dendritic locations. Indeed, in the primary sensory areas of the neocortex, feedback from higher-order areas arrives in the distal apical dendrites of pyramidal neurons (*Manita et al., 2015*; *Budd, 1998*; *Spratling, 2002*), which are electrotonically very distant from the basal dendrites where feedforward sensory information is received (*Larkum et al., 1999*; *2007*; *2009*). Thus, as has been noted by previous authors (*Körding and König, 2001*; *Spratling, 2002*; *Spratling and Johnson, 2006*), the anatomy of pyramidal neurons may actually provide the segregation of feedforward and feedback information required to calculate local error signals and perform credit assignment in biological neural networks.

Here, we show how deep learning can be implemented if neurons in hidden layers contain segregated 'basal' and 'apical' dendritic compartments for integrating feedforward and feedback signals separately (*Figure 2B*). Our model builds on previous neural networks research (*Lee et al., 2015*; *Lillicrap et al., 2016*) as well as computational studies of supervised learning in multi-compartment

neurons (*Urbanczik and Senn, 2014*; *Körding and König, 2001*; *Spratling and Johnson, 2006*). Importantly, we use the distinct basal and apical compartments in our neurons to integrate feedback signals separately from feedforward signals. With this, we build a local error signal for each hidden layer that ensures appropriate credit assignment. We demonstrate that even with random synaptic weights for feedback into the apical compartment, our algorithm can coordinate learning to achieve classification of the MNIST database of hand-written digits that is better than that which can be achieved with a single layer network. Furthermore, we show that our algorithm allows the network to take advantage of multi-layer structures to build hierarchical, abstract representations, one of the hallmarks of deep learning (*LeCun et al., 2015*). Our results demonstrate that deep learning can be implemented in a biologically feasible manner if feedforward and feedback signals are received at electrotonically segregated dendrites, as is the case in the mammalian neocortex.

## Results

### A network architecture with segregated dendritic compartments

Deep supervised learning with local weight updates requires that each neuron receive signals that can be used to determine its 'credit' for the final behavioral output. We explored the idea that the cortico-cortical feedback signals to pyramidal cells could provide the required information for credit assignment. In particular, we were inspired by four observations from both machine learning and biology:

1. Current solutions to credit assignment without weight transport require segregated feedforward and feedback signals (*Lee et al., 2015*; *Lillicrap et al., 2016*).
2. In the neocortex, feedforward sensory information and higher-order cortico-cortical feedback are largely received by distinct dendritic compartments, namely the basal dendrites and distal apical dendrites, respectively (*Spratling, 2002*; *Budd, 1998*).
3. The distal apical dendrites of pyramidal neurons are electrotonically distant from the soma, and apical communication to the soma depends on active propagation through the apical dendritic shaft, which is predominantly driven by voltage-gated calcium channels. Due to the dynamics of voltage-gated calcium channels these non-linear, active events in the apical shaft generate prolonged upswings in the membrane potential, known as 'plateau potentials', which can drive burst firing at the soma (*Larkum et al., 1999*; *2009*).
4. Plateau potentials driven by apical activity can guide plasticity in pyramidal neurons *in vivo* (*Bittner et al., 2015*; *Bittner et al., 2017*).

With these considerations in mind, we hypothesized that the computations required for credit assignment could be achieved without separate pathways for feedback signals. Instead, they could be achieved by having two distinct dendritic compartments in each hidden layer neuron: a 'basal' compartment, strongly coupled to the soma for integrating bottom-up sensory information, and an 'apical' compartment for integrating top-down feedback in order calculate credit assignment and drive synaptic plasticity via 'plateau potentials' (*Bittner et al., 2015*; *Bittner et al., 2017*) (*Figure 3A*).

As an initial test of this concept we built a network with a single hidden layer. Although this network is not very 'deep', even a single hidden layer can improve performance over a one-layer architecture if the learning algorithm solves the credit assignment problem (*Bengio and LeCun, 2007*; *Lillicrap et al., 2016*). Hence, we wanted to initially determine whether our network could take advantage of a hidden layer to reduce error at the output layer.

The network architecture is illustrated in *Figure 3B*. An image from the MNIST data set is used to set the spike rates of $\ell = 784$ Poisson point-process neurons in the input layer (one neuron per image pixel, rates-of-fire determined by pixel intensity). These project to a hidden layer with $m = 500$ neurons. The neurons in the hidden layer (which we index with a '0') are composed of three distinct compartments with their own voltages: the apical compartments (with voltages described by the vector $\boldsymbol{V}^{0a}(t) = [V_1^{0a}(t), ..., V_m^{0a}(t)]$), the basal compartments (with voltages $\boldsymbol{V}^{0b}(t) = [V_1^{0b}(t), ..., V_m^{0b}(t)]$), and the somatic compartments (with voltages $\boldsymbol{V}^0(t) = [V_1^0(t), ..., V_m^0(t)]$). (*Note*: for notational clarity, all vectors and matrices in the paper are in boldface.) The voltage of the $i^{th}$ neuron in the hidden layer is updated according to:

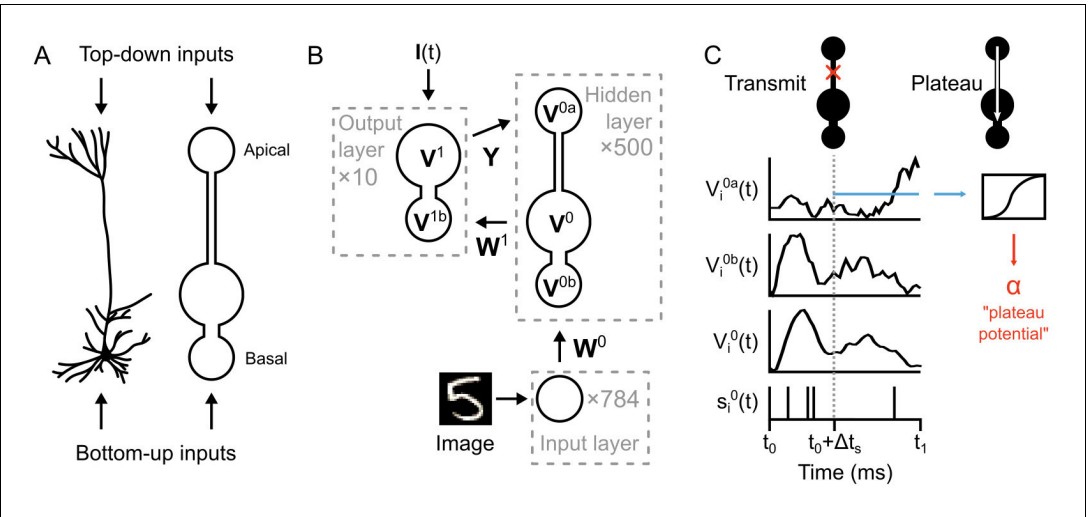

**Figure 3.** Illustration of a multi-compartment neural network model for deep learning. (**A**) *Left*: Reconstruction of a real pyramidal neuron from layer five mouse primary visual cortex. *Right*: Illustration of our simplified pyramidal neuron model. The model consists of a somatic compartment, plus two distinct dendritic compartments (apical and basal). As in real pyramidal neurons, top-down inputs project to the apical compartment while bottom-up inputs project to the basal compartment. (**B**) Diagram of network architecture. An image is used to drive spiking input units which project to the hidden layer basal compartments through weights $W^0$. Hidden layer somata project to the output layer dendritic compartment through weights $W^1$. Feedback from the output layer somata is sent back to the hidden layer apical compartments through weights $Y$. The variables for the voltages in each of the compartments are shown. The number of neurons used in each layer is shown in gray. (**C**) Illustration of transmit vs. plateau computations. *Left*: In the transmit computation, the network dynamics are updated at each time-step, and the apical dendrite is segregated by a low value for $g_a$, making the network effectively feedforward. Here, the voltages of each of the compartments are shown for one run of the network. The spiking output of the soma is also shown. Note that the somatic voltage and spiking track the basal voltage, and ignore the apical voltage. However, the apical dendrite does receive feedback, and this is used to drive its voltage. After a period of $\Delta t_s$ to allow for settling of the dynamics, the average apical voltage is calculated (shown here as a blue line). *Right*: The average apical voltage is then used to calculate an apical plateau potential, which is equal to the nonlinearity $\sigma(\cdot)$ applied to the average apical voltage.
DOI: https://doi.org/10.7554/eLife.22901.005

$$\tau\frac{dV_i^0(t)}{dt} = -V_i^0(t) + \frac{g_b}{g_l}(V_i^{0b}(t) - V_i^0(t)) + \frac{g_a}{g_l}(V_i^{0a}(t) - V_i^0(t)) \tag{1}$$

where $g_l$, $g_b$ and $g_a$ represent the leak conductance, the conductance from the basal dendrites, and the conductance from the apical dendrites, respectively, and $\tau = C_m/g_l$ where $C_m$ is the membrane capacitance (see Materials and methods, *Equation (16)*). For mathematical simplicity we assume in our simulations a resting membrane potential of 0 mV (this value does not affect the results). We implement electrotonic segregation in the model by altering the $g_a$ value—low values for $g_a$ lead to electrotonically segregated apical dendrites. In the initial set of simulations we set $g_a = 0$, which effectively makes it a feed-forward network, but we relax this condition in later simulations.

We treat the voltages in the dendritic compartments simply as weighted sums of the incoming spike trains. Hence, for the $i^{th}$ hidden layer neuron:

$$\begin{aligned} V_i^{0b}(t) &= \sum_{j=1}^{\ell} W_{ij}^0 s_j^{\text{input}}(t) + b_i^0 \\ V_i^{0a}(t) &= \sum_{j=1}^{n} Y_{ij} s_j^1(t) \end{aligned} \tag{2}$$

where $W_{ij}^0$ and $Y_{ij}$ are synaptic weights from the input layer and the output layer, respectively, $b_i^0$ is a

bias term, and $s^{\text{input}}$ and $s^1$ are the filtered spike trains of the input layer and output layer neurons, respectively. (Note: the spike trains are convolved with an exponential kernel to mimic postsynaptic potentials, see Materials and methods *Equation (11)*.)

The somatic compartments generate spikes using Poisson processes. The instantaneous rates of these processes are described by the vector $\boldsymbol{\phi}^0(t) = [\phi_1^0(t), ..., \phi_m^0(t)]$, which is in units of spikes/s or Hz. These rates-of-fire are determined by a non-linear sigmoid function, $\sigma(\cdot)$, applied to the somatic voltages, that is for the $i^{th}$ hidden layer neuron:

$$
\begin{aligned}
\phi_i^0(t) &= \phi_{\max}\sigma(V_i^0(t)) \\
&= \phi_{\max}\frac{1}{1+e^{-V_i^0(t)}}
\end{aligned}
\tag{3}
$$

where $\phi_{\max}$ is the maximum rate-of-fire for the neurons.

The output layer (which we index here with a '1') contains $n = 10$ two-compartment neurons (one for each image category), similar to those used in a previous model of dendritic prediction learning (*Urbanczik and Senn, 2014*). The output layer dendritic voltages ($\boldsymbol{V}^{1b}(t) = [V_1^{1b}(t), ..., V_n^{1b}(t)]$) and somatic voltages ($\boldsymbol{V}^1(t) = [V_1^1(t), ..., V_n^1(t)]$) are updated in a similar manner to the hidden layer basal compartment and soma:

$$
\begin{aligned}
\tau\frac{dV_i^1(t)}{dt} &= -V_i^1(t) + \frac{g_d}{g_l}(V_i^{1b}(t) - V_i^1(t)) + I_i(t) \\
V_i^{1b}(t) &= \sum_{j=1}^{\ell} W_{ij}^1 s_j^0(t) + b_i^1
\end{aligned}
\tag{4}
$$

where $W_{ij}^1$ are synaptic weights from the hidden layer, $s^0$ are the filtered spike trains of the hidden layer neurons (see *Equation (11)*), $g_l$ is the leak conductance, $g_d$ is the conductance from the dendrites, and $\tau$ is given by *Equation (16)*. In addition to the absence of an apical compartment, the other salient difference between the output layer neurons and the hidden layer neurons is the presence of the term $I_i(t)$, which is a teaching signal that can be used to force the output layer to the correct answer. Whether any such teaching signals exist in the real brain is unknown, though there is evidence that animals can represent desired behavioral outputs with internal goal representations (*Gadagkar et al., 2016*). (See below, and Materials and methods, *Equations (19) and (20)* for more details on the teaching signal).

In our model, there are two different types of computation that occur in the hidden layer neurons: 'transmit' and 'plateau'. The transmit computations are standard numerical integration of the simulation, with voltages evolving according to *Equation (1)*, and with the apical compartment electrotonically segregated from the soma (depending on $g_a$) (*Figure 3C*, left). In contrast, the plateau computations *do not* involve numerical integration with *Equation (1)*. Instead, the apical voltage is averaged over the most recent 20–30 ms period and the sigmoid non-linearity is applied to it, giving us 'plateau potentials' in the hidden layer neurons (we indicate plateau potentials with $\alpha$, see *Equation (5)* below, and *Figure 3C*, right). The intention behind this design was to mimic the non-linear transmission from the apical dendrites to the soma that occurs during a plateau potential driven by calcium spikes in the apical dendritic shaft (*Larkum et al., 1999*), but in the simplest, most abstract formulation possible.

Importantly, plateau potentials in our simulations are single numeric values (one per hidden layer neuron) that can be used for credit assignment. We do not use them to alter the network dynamics. When they occur, they are calculated, transmitted to the basal dendrite instantaneously, and then stored temporarily (0–60 ms) for calculating synaptic weight updates.

## Calculating credit assignment signals with feedback driven plateau potentials

To train the network we alternate between two phases. First, during the 'forward' phase we present an image to the input layer without any teaching current at the output layer ($I(t)_i = 0, \forall i$). The forward phase occurs between times $t_0$ to $t_1$. At $t_1$ a plateau potential is calculated in all the hidden layer neurons ($\boldsymbol{\alpha}^f = [\alpha_1^f, ..., \alpha_m^f]$) and the 'target' phase begins. During this phase, which lasts until $t_2$,

the image continues to drive the input layer, but now the output layer also receives teaching current. The teaching current forces the correct output neuron to its max firing rate and all the others to silence. For example, if an image of a '9' is presented, then over the time period $t_1$-$t_2$ the '9' neuron in the output layer fires at max, while the other neurons are silent (*Figure 4A*). At $t_2$ another set of plateau potentials ($\boldsymbol{\alpha}^t = [\alpha_1^t, ..., \alpha_m^t]$) are calculated in the hidden layer neurons. The result is that we have plateau potentials in the hidden layer neurons for both the end of the forward phase ($\boldsymbol{\alpha}^f$) and the end of the target phase ($\boldsymbol{\alpha}^t$), which are calculated as:

$$\begin{aligned}
\alpha_i^f &= \sigma\left(\frac{1}{\Delta t_1}\int_{t_1-\Delta t_1}^{t_1} V_i^{0a}(t)dt\right) \\
\alpha_i^t &= \sigma\left(\frac{1}{\Delta t_2}\int_{t_2-\Delta t_2}^{t_2} V_i^{0a}(t)dt\right)
\end{aligned} \tag{5}$$

where $\Delta t_s$ is a time delay used to allow the network dynamics to settle before integrating the plateau, and $\Delta t_i = t_i - (t_{i-1} + \Delta t_s)$ (see Materials and methods, *Equation (22)* and *Figure 4A*).

Similar to how targets are used in deep supervised learning (*LeCun et al., 2015*), the goal of learning in our network is to make the network dynamics during the forward phase converge to the same output activity pattern as exists in the target phase. Put another way, in the absence of the teaching signal, we want the activity at the output layer to be the same as that which would exist with the teaching signal, so that the network can give appropriate outputs without any guidance. To do this, we initialize all the weight matrices with random weights, then we train the weight matrices $W^0$ and $W^1$ using stochastic gradient descent on local loss functions for the hidden and output layers, respectively (see below). These weight updates occur at the end of every target phase, that is the synapses are not updated during transmission. Like *Lillicrap et al. (2016)*, we leave the weight

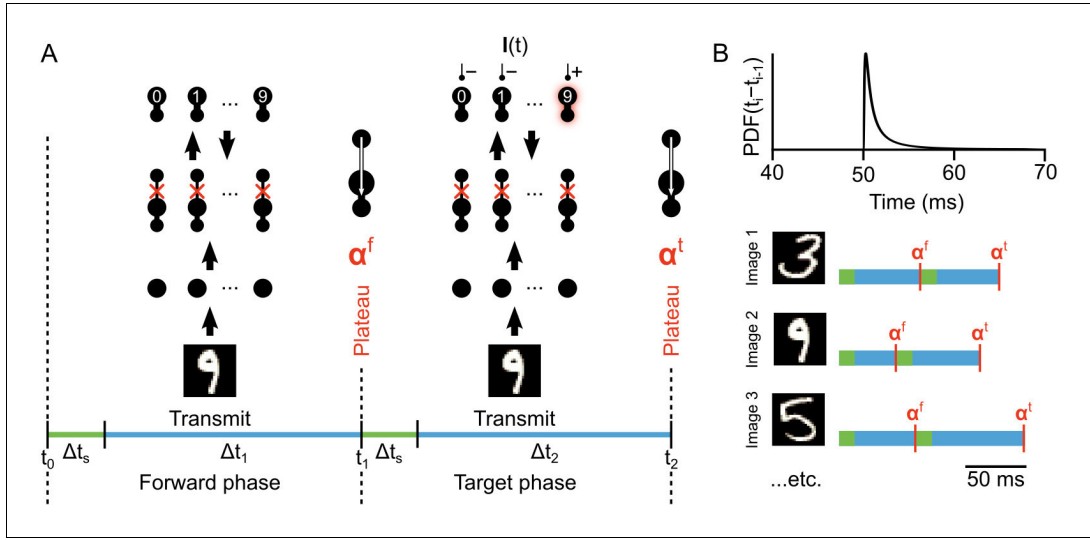

**Figure 4.** Illustration of network phases for learning. (A) Illustration of the sequence of network phases that occur for each training example. The network undergoes a forward phase where $I_i(t) = 0$, $\forall i$ and a target phase where $I_i(t)$ causes any given neuron $i$ to fire at max-rate or be silent, depending on whether it is the correct category of the current input image. In this illustration, an image of a '9' is being presented, so the '9' unit at the output layer is activated and the other output neurons are inhibited and silent. At the end of the forward phase the set of plateau potentials $\boldsymbol{\alpha}^f$ are calculated, and at the end of the target phase the set of plateau potentials $\boldsymbol{\alpha}^t$ are calculated. (B) Illustration of phase length sampling. Each phase length is sampled stochastically. In other words, for each training image, the lengths of forward and target phases (shown as blue bar pairs, where bar length represents phase length) are randomly drawn from a shifted inverse Gaussian distribution with a minimum of 50 ms.

DOI: https://doi.org/10.7554/eLife.22901.006

matrix $\boldsymbol{Y}$ fixed in its initial random configuration. When we update the synapses in the network we use the plateau potential values $\boldsymbol{\alpha}^f$ and $\boldsymbol{\alpha}^t$ to determine appropriate credit assignment (see below).

The network is simulated in near continuous-time (except that each plateau is considered to be instantaneous), and the temporal intervals between plateaus are randomly sampled from an inverse Gaussian distribution (*Figure 4B*, top). As such, the specific amount of time that the network is presented with each image and teaching signal is stochastic, though usually somewhere between 50–60 ms of simulated time (*Figure 4B*, bottom). This stochasticity was not necessary, but it demonstrates that although the system operates in phases, the specific length of the phases is not important as long as they are sufficiently long to permit integration (see Lemma 1). In the data presented in this paper, all 60,000 images in the MNIST training set were presented to the network one at a time, and each exposure to the full set of images was considered an 'epoch' of training. At the end of each epoch, the network's classification error rate on a separate set of 10,000 test images was assessed with a single forward phase for each image (see Materials and methods). The network's classification was judged by which output neuron had the highest average firing rate during these test image forward phases.

It is important to note that there are many aspects of this design that are not physiologically accurate. Most notably, stochastic generation of plateau potentials across a population is not an accurate reflection of how real pyramidal neurons operate, since apical calcium spikes are determined by a number of concrete physiological factors in individual cells, including back-propagating action potentials, spike-timing and inhibitory inputs (*Larkum et al., 1999*, *2007*, *2009*). However, we note that calcium spikes in the apical dendrites can be prevented from occurring via the activity of distal dendrite targeting inhibitory interneurons (*Murayama et al., 2009*), which can synchronize pyramidal activity (*Hilscher et al., 2017*). Furthermore, distal dendrite targeting interneurons can themselves can be rapidly inhibited in response to temporally precise neuromodulatory inputs (*Pi et al., 2013*; *Pfeffer et al., 2013*; *Karnani et al., 2016*; *Hangya et al., 2015*; *Brombas et al., 2014*). Therefore, it is entirely plausible that neocortical micro-circuits would generate synchronized plateaus/bursts at punctuated periods of time in response to disinhibition of the apical dendrites governed by neuromodulatory signals that determine 'phases' of processing. Alternatively, oscillations in population activity could provide a mechanism for promoting alternating phases of processing and synaptic plasticity (*Buzsáki and Draguhn, 2004*). But, complete synchrony of plateaus in our hidden layer neurons is not actually critical to our algorithm—only the temporal relationship between the plateaus and the teaching signal is critical. This relationship itself is arguably plausible given the role of neuromodulatory inputs in dis-inhibiting the distal dendrites of pyramidal neurons (*Karnani et al., 2016*; *Brombas et al., 2014*). Of course, we are engaged in a great deal of speculation here. But, the point is that our model utilizes anatomical and functional motifs that are loosely analogous to what is observed in the neocortex. Importantly for the present study, the key issue is the use of segregated dendrites which permit an effective feed-forward dynamic, punctuated by feedback driven plateau potentials to solve the credit assignment problem.

## Co-ordinating optimization across layers with feedback to apical dendrites

To solve the credit assignment problem without using weight transport, we had to define local error signals, or 'loss functions', for the hidden layer and output layer that somehow took into account the impact that each hidden layer neuron has on the output of the network. In other words, we only want to update a hidden layer synapse in a manner that will help us make the forward phase activity at the output layer more similar to the target phase activity. To begin, we define the target firing rates for the output neurons, $\boldsymbol{\phi}^{1*} = [\phi_1^{1*}, ..., \phi_n^{1*}]$, to be their average firing rates during the target phase:

$$\begin{aligned} \phi_i^{1*} &= \overline{\phi_i^1}^t \\ &= \frac{1}{\Delta t_2} \int_{t_1+\Delta t_s}^{t_2} \phi_i^1(t)dt \end{aligned} \quad (6)$$

(Throughout the paper, we use $\phi^*$ to denote a target firing rate and $\overline{\phi}$ to denote a firing rate

averaged over time.) We then define a loss function at the output layer using this target, by taking the difference between the average forward phase activity and the target:

$$
\begin{aligned}
L^1 &\approx ||\boldsymbol{\phi}^{1*} - \overline{\boldsymbol{\phi}^{1^f}}||_2^2 \\
&= ||\overline{\boldsymbol{\phi}^{1^t}} - \overline{\boldsymbol{\phi}^{1^f}}||_2^2 \\
&= \left|\left| \frac{1}{\Delta t_2} \int_{t_1+\Delta t_s}^{t_2} \boldsymbol{\phi}^1(t)dt - \frac{1}{\Delta t_1} \int_{t_0+\Delta t_s}^{t_1} \boldsymbol{\phi}^1(t)dt \right|\right|_2^2
\end{aligned}
\tag{7}
$$

(Note: the true loss function we use is slightly more complex than the one formulated here, hence the $\approx$ symbol in *Equation (7)*, but this formulation is roughly correct and easier to interpret. See Materials and methods, *Equation (23)* for the exact formulation.) This loss function is zero only when the average firing rates of the output neurons during the forward phase equals their target, that is the average firing rates during the target phase. Thus, the closer $L^1$ is to zero, the more the network's output for an image matches the output activity pattern imposed by the teaching signal, $I(t)$.

Effective credit assignment is achieved when changing the hidden layer synapses is guaranteed to reduce $L^1$. To obtain this guarantee, we defined a set of target firing rates for the hidden layer neurons that uses the information contained in the plateau potentials. Specifically, in a similar manner to *Lee et al., 2015*, we define the target firing rates for the hidden layer neurons, $\boldsymbol{\phi}^{0*} = [\phi_1^{0*}, ..., \phi_m^{0*}]$, to be:

$$
\phi_i^{0*} = \overline{\phi_i^{0^f}} + \alpha_i^t - \alpha_i^f
\tag{8}
$$

where $\alpha_i^t$ and $\alpha_i^f$ are the plateaus defined in *Equation (5)*. As with the output layer, we define the loss function for the hidden layer to be the difference between the target firing rate and the average firing rate during the forward phase:

$$
\begin{aligned}
L^0 &\approx ||\boldsymbol{\phi}^{0*} - \overline{\boldsymbol{\phi}^{0^f}}||_2^2 \\
&= ||\overline{\boldsymbol{\phi}^{0^f}} + \boldsymbol{\alpha}^t - \boldsymbol{\alpha}_i^f - \overline{\boldsymbol{\phi}^{0^f}}||_2^2 \\
&= ||\boldsymbol{\alpha}^t - \boldsymbol{\alpha}^f||_2^2
\end{aligned}
\tag{9}
$$

(Again, note the use of the $\approx$ symbol, see *Equation (30)* for the exact formulation.) This loss function is zero only when the plateau at the end of the forward phase equals the plateau at the end of the target phase. Since the plateau potentials integrate the top-down feedback (see *Equation (5)*), we know that the hidden layer loss function, $L^0$, is zero if the output layer loss function, $L^1$, is zero. Moreover, we can show that these loss functions provide a broader guarantee that, under certain conditions, if $L^0$ is reduced, then on average, $L^1$ will also be reduced (see Theorem 1). This provides our assurance of credit assignment: we know that the ultimate goal of learning (reducing $L^1$) can be achieved by updating the synaptic weights at the hidden layer to reduce the local loss function $L^0$ (*Figure 5A*). We do this using stochastic gradient descent at the end of every target phase:

$$
\begin{aligned}
\Delta \boldsymbol{W}^1 &= -\eta_0 \frac{\partial L^1}{\partial \boldsymbol{W}^1} \\
\Delta \boldsymbol{W}^0 &= -\eta_1 \frac{\partial L^0}{\partial \boldsymbol{W}^0}
\end{aligned}
\tag{10}
$$

where $\eta_i$ and $\Delta \boldsymbol{W}^i$ refer to the learning rate and update term for weight matrix $\boldsymbol{W}^i$ (see Materials and methods, *Equations (28), (29), (33) and (35)* for details of the weight update procedures). Performing gradient descent on $L^1$ results in a relatively straight-forward delta rule update for $\boldsymbol{W}^1$ (see *Equation (29)*). The weight update for the hidden layer weights, $\boldsymbol{W}^0$, is similar, except for the presence of the difference between the two plateau potentials $\boldsymbol{\alpha}^t - \boldsymbol{\alpha}^f$ (see *Equation (35)*). Importantly, given the way in which we defined the loss functions, as the hidden layer reduces $L^0$ by updating $\boldsymbol{W}^0$, $L^1$ should also be reduced, that is hidden layer learning should imply output layer learning, thereby utilizing the multi-layer architecture.

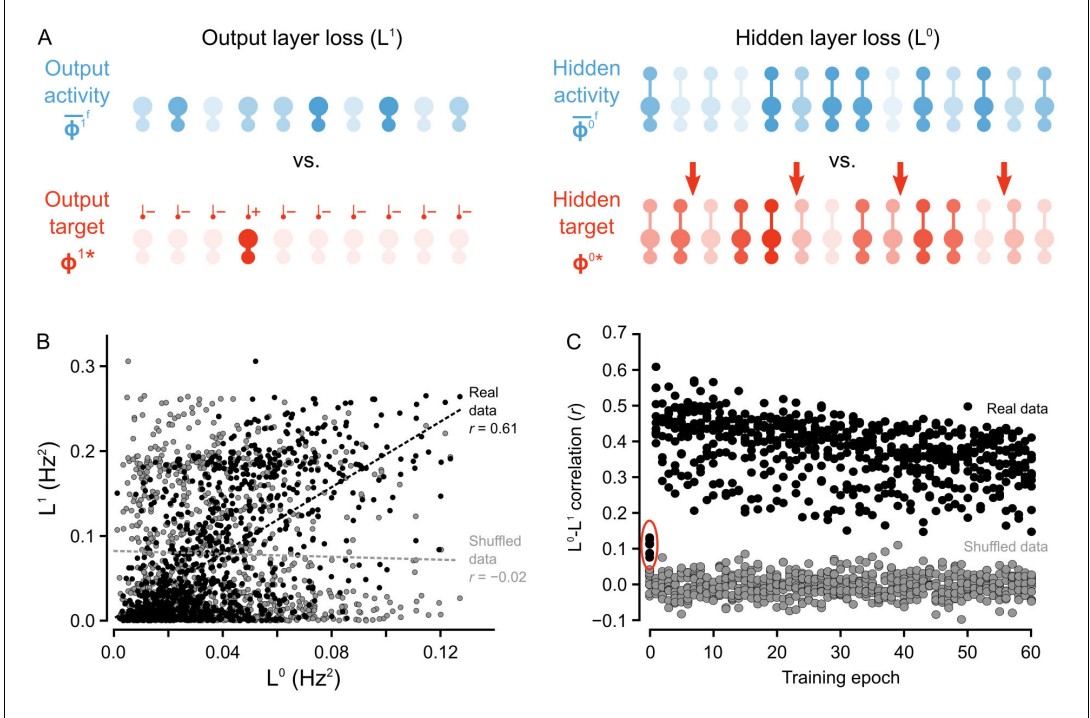

**Figure 5.** Co-ordinated errors between the output and hidden layers. (**A**) Illustration of output loss function ($L^1$) and local hidden loss function ($L^0$). For a given test example shown to the network in a forward phase, the output layer loss is defined as the squared norm of the difference between target firing rates $\phi^{1*}$ and the average firing rate during the forward phases of the output units. Hidden layer loss is defined similarly, except the target is $\phi^{0*}$ (as defined in the text). (**B**) Plot of $L^1$ vs. $L^0$ for all of the '2' images after one epoch of training. There is a strong correlation between hidden layer loss and output layer loss (real data, black), as opposed to when output and hidden loss values were randomly paired (shuffled data, gray). (**C**) Plot of correlation between hidden layer loss and output layer loss across training for each category of images (each dot represents one category). The correlation is significantly higher in the real data than the shuffled data throughout training. Note also that the correlation is much lower on the first epoch of training (red oval), suggesting that the conditions for credit assignment are still developing during the first epoch.
DOI: https://doi.org/10.7554/eLife.22901.007

The following source data and figure supplement are available for figure 5:

**Source data 1.** Fig_5B.csv.
DOI: https://doi.org/10.7554/eLife.22901.009

**Figure supplement 1.** Weight alignment during first epoch of training.
DOI: https://doi.org/10.7554/eLife.22901.008

To test that we were successful in credit assignment with this design, and to provide empirical support for the proof of Theorem 1, we compared the loss function at the hidden layer, $L^0$, to the output layer loss function, $L^1$, across all of the image presentations to the network. We observed that, generally, whenever the hidden layer loss was low, the output layer loss was also low. For example, when we consider the loss for the set of '2' images presented to the network during the second epoch, there was a Pearson correlation coefficient between $L^0$ and $L^1$ of $r = 0.61$, which was much higher than what was observed for shuffled data, wherein output and hidden activities were randomly paired (*Figure 5B*). Furthermore, these correlations were observed across all epochs of training, with most correlation coefficients for the hidden and output loss functions falling between $r = 0.2 - 0.6$, which was, again, much higher than the correlations observed for shuffled data (*Figure 5C*).

Interestingly, the correlations between $L^0$ and $L^1$ were smaller on the first epoch of training (see data in red oval *Figure 5C*). This suggests that the guarantee of coordination between $L^0$ and $L^1$ only comes into full effect once the network has engaged in some learning. Therefore, we inspected whether the conditions on the synaptic matrices that are assumed in the proof of Theorem 1 were, in fact, being met. More precisely, the proof assumes that the feedforward and feedback synaptic

matrices ($W^1$ and $Y$, respectively) produce forward and backward transformations between the output and hidden layer whose Jacobians are approximate inverses of each other (see Proof of Theorem 1). Since we begin learning with random matrices, this condition is almost definitely *not* met at the start of training. But, we found that the network learned to meet this condition. Inspection of $W^1$ and $Y$ showed that during the first epoch, the Jacobians of the forward and backwards functions became approximate inverses of each other (*Figure 5—figure supplement 1*). Since $Y$ is frozen, this means that during the first few image presentations $W^1$ was being updated to have its Jacobian come closer to the inverse of $Y$'s Jacobian. Put another way, the network was *learning to do credit assignment*. We have yet to resolve exactly why this happens, though the result is very similar to the findings of *Lillicrap et al. (2016)*, where a proof is provided for the linear case. Intuitively, though, the reason is likely the interaction between $W^1$ and $W^0$: as $W^0$ gets updated, the hidden layer learns to group stimuli based on the feedback sent through $Y$. So, for $W^1$ to transform the hidden layer activity into the correct output layer activity, $W^1$ must become more like the inverse of $Y$, which would also make the Jacobian of $W^1$ more like the inverse of $Y$'s Jacobian (due to the inverse function theorem). However, a complete, formal explanation for this phenomenon is still missing, and the the issue of weight alignment deserves additional investigation *Lillicrap et al. (2016)*. From a biological perspective, it also suggests that very early development may involve a period of learning how to assign credit appropriately. Altogether, our model demonstrates that deep learning using random feedback weights is a general phenomenon, and one which can be implemented using segregated dendrites to keep forward information separate from feedback signals used for credit assignment.

## Deep learning with segregated dendrites

Given our finding that the network was successfully assigning credit for the output error to the hidden layer neurons, we had reason to believe that our network with local weight-updates would exhibit deep learning, that is an ability to take advantage of a multi-layer structure (*Bengio and LeCun, 2007*). To test this, we examined the effects of including hidden layers. If deep learning is indeed operational in the network, then the inclusion of hidden layers should improve the ability of the network to classify images.

We built three different versions of the network (*Figure 6A*). The first was a network that had no hidden layer, that is the input neurons projected directly to the output neurons. The second was the network illustrated in *Figure 3B*, with a single hidden layer. The third contained two hidden layers, with the output layer projecting directly back to both hidden layers. This direct projection allowed us to build our local targets for each hidden layer using the plateaus driven by the output layer, thereby avoiding a 'backward pass' through the entire network as has been used in other models (*Lillicrap et al., 2016*; *Lee et al., 2015*; *Liao et al., 2015*). We trained each network on the 60,000 MNIST training images for 60 epochs, and recorded the percentage of images in the 10,000 image test set that were incorrectly classified. The network with no hidden layers rapidly learned to classify the images, but it also rapidly hit an asymptote at an average error rate of 8.3% (*Figure 6B*, gray line). In contrast, the network with one hidden layer did not exhibit a rapid convergence to an asymptote in its error rate. Instead, it continued to improve throughout all 60 epochs, achieving an average error rate of 4.1% by the 60$^{th}$ epoch (*Figure 6B*, blue line). Similar results were obtained when we loosened the synchrony constraints and instead allowed each hidden layer neuron to engage in plateau potentials at different times (*Figure 6—figure supplement 1*). This demonstrates that strict synchrony in the plateau potentials is not required. But, our target definitions do require two different plateau potentials separated by the teaching signal input, which mandates some temporal control of plateau potentials in the system.

Interestingly, we found that the addition of a second hidden layer further improved learning. The network with two hidden layers learned more rapidly than the network with one hidden layer and achieved an average error rate of 3.2% on the test images by the 60$^{th}$ epoch, also without hitting a clear asymptote in learning (*Figure 6B*, red line). However, it should be noted that additional hidden layers beyond two did not significantly improve the error rate (data not shown), which suggests that our particular algorithm could not be used to construct very deep networks as is. Nonetheless, our network was clearly able to take advantage of multi-layer architectures to improve its learning, which is the key feature of deep learning (*Bengio and LeCun, 2007*; *LeCun et al., 2015*).

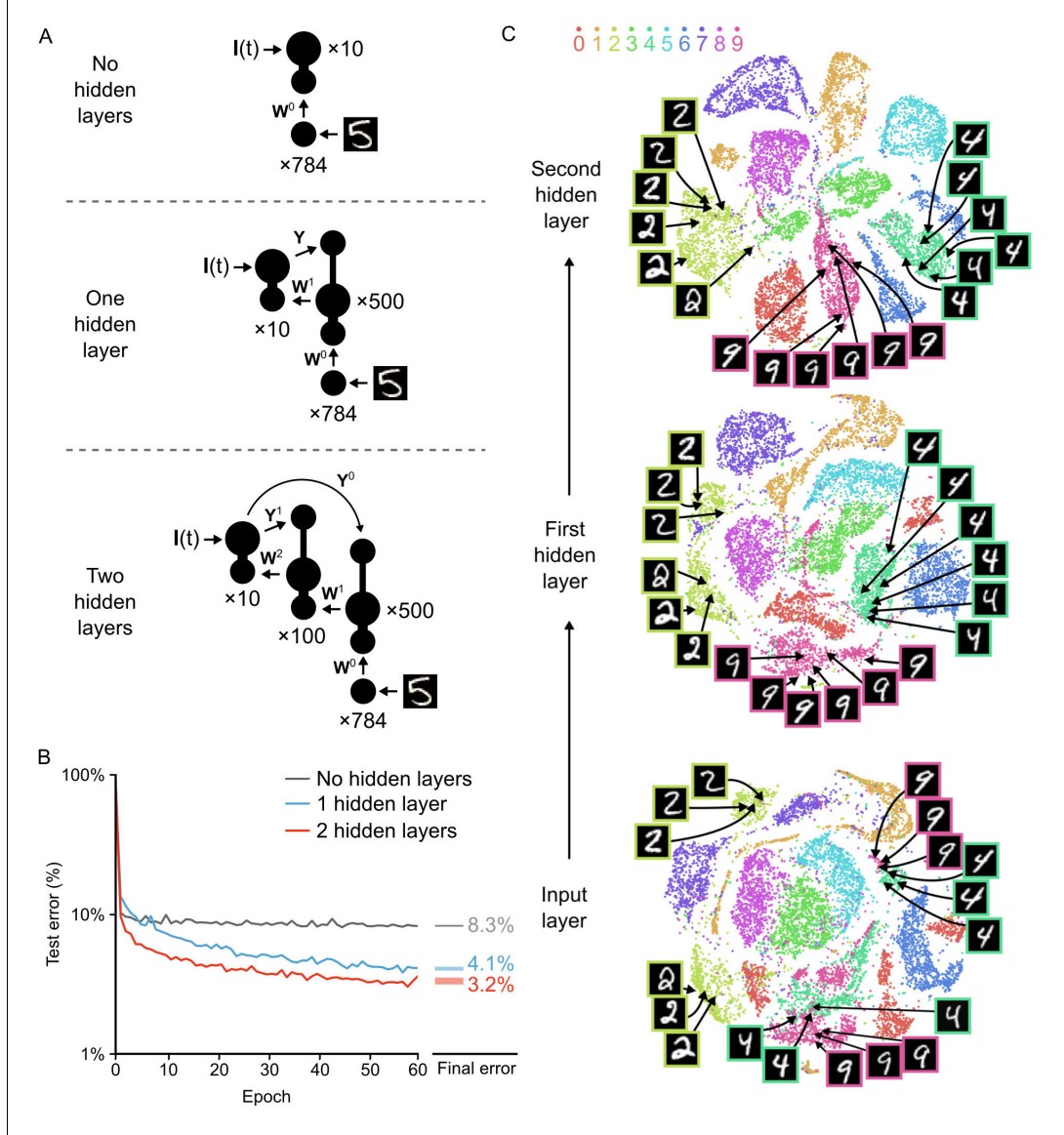

**Figure 6.** Improvement of learning with hidden layers. (**A**) Illustration of the three networks used in the simulations. *Top*: a shallow network with only an input layer and an output layer. *Middle*: a network with one hidden layer. *Bottom*: a network with two hidden layers. Both hidden layers receive feedback from the output layer, but through separate synaptic connections with random weights $Y^0$ and $Y^1$. (**B**) Plot of test error (measured on 10,000 MNIST images not used for training) across 60 epochs of training, for all three networks described in A. The networks with hidden layers exhibit deep learning, because hidden layers decrease the test error. *Right*: Spreads (min – max) of the results of repeated weight tests ($n = 20$) after 60 epochs for each of the networks. Percentages indicate means (two-tailed t-test, 1-layer vs. 2-layer: $t_{38} = 197.11$, $p = 2.5 \times 10^{-58}$; 1-layer vs. 3-layer: $t_{38} = 238.26$, $p = 1.9 \times 10^{-61}$; 2-layer vs. 3-layer: $t_{38} = 42.99$, $p = 2.3 \times 10^{-33}$, Bonferroni correction for multiple comparisons). (**C**) Results of t-SNE dimensionality reduction applied to the activity patterns of the first three layers of a two hidden layer network (after 60 epochs of training). Each data point corresponds to a test image shown to the network. Points are color-coded according to the digit they represent. Moving up through the network, images from identical categories are clustered closer together and separated from images of different categories. Thus the hidden layers learn increasingly abstract representations of digit categories.

DOI: https://doi.org/10.7554/eLife.22901.010

The following source data and figure supplement are available for figure 6:

**Source data 1.** Fig_6B_errors.csv.
DOI: https://doi.org/10.7554/eLife.22901.012

**Figure supplement 1.** Learning with stochastic plateau times.
DOI: https://doi.org/10.7554/eLife.22901.011

Another key feature of deep learning is the ability to generate representations in the higher layers of a network that capture task-relevant information while discarding sensory details (*LeCun et al., 2015*; *Mnih et al., 2015*). To examine whether our network exhibited this type of abstraction, we used the t-Distributed Stochastic Neighbor Embedding algorithm (t-SNE). The t-SNE algorithm reduces the dimensionality of data while preserving local structure and non-linear manifolds that exist in high-dimensional space, thereby allowing accurate visualization of the structure of high-dimensional data (*Maaten and Hinton, 2008*). We applied t-SNE to the activity patterns at each layer of the two hidden layer network for all of the images in the test set after 60 epochs of training. At the input level, there was already some clustering of images based on their categories. However, the clusters were quite messy, with different categories showing outliers, several clusters, or merged clusters (*Figure 6C*, bottom). For example, the '2' digits in the input layer exhibited two distinct clusters separated by a cluster of '7's: one cluster contained '2's with a loop and one contained '2's without a loop. Similarly, there were two distinct clusters of '4's and '9's that were very close to each other, with one pair for digits on a pronounced slant and one for straight digits (*Figure 6C*, bottom, example images). Thus, although there is built-in structure to the categories of the MNIST dataset, there are a number of low-level features that do not respect category boundaries. In contrast, at the first hidden layer, the activity patterns were much cleaner, with far fewer outliers and split/merged clusters (*Figure 6C*, middle). For example, the two separate '2' digit clusters were much closer to each other and were now only separated by a very small cluster of '7's. Likewise, the '9' and '4' clusters were now distinct and no longer split based on the slant of the digit. Interestingly, when we examined the activity patterns at the second hidden layer, the categories were even better segregated, with only a little bit of splitting or merging of category clusters (*Figure 6C*, top). Therefore, the network had learned to develop representations in the hidden layers wherein the categories were very distinct and low-level features unrelated to the categories were largely ignored. This abstract representation is likely to be key to the improved error rate in the two hidden layer network. Altogether, our data demonstrates that our network with segregated dendritic compartments can engage in deep learning.

## Coordinated local learning mimics backpropagation of error

The backpropagation of error algorithm (*Rumelhart et al., 1986*) is still the primary learning algorithm used for deep supervised learning in artificial neural networks (*LeCun et al., 2015*). Previous work has shown that learning with random feedback weights can actually match the synaptic weight updates specified by the backpropagation algorithm after a few epochs of training (*Lillicrap et al., 2016*). This fascinating observation suggests that deep learning with random feedback weights is not completely distinct from backpropagation of error, but rather, networks with random feedback connections learn to approximate credit assignment as it is done in backpropagation (*Lillicrap et al., 2016*). Hence, we were curious as to whether or not our network was, in fact, learning to approximate the synaptic weight updates prescribed by backpropagation. To test this, we trained our one hidden layer network as before, but now, in addition to calculating the vector of hidden layer synaptic weight updates specified by our local learning rule ($\Delta W^0$ in *Equation (10)*), we also calculated the vector of hidden layer synaptic weight updates that would be specified by non-locally backpropagating the error from the output layer, ($\Delta W^0_{BP}$). We then calculated the angle between these two alternative weight updates. In a very high-dimensional space, any two independent vectors will be roughly orthogonal to each other (i.e. $\Delta W^0 \angle \Delta W^0_{BP} \approx 90°$). If the two synaptic weight update vectors are *not* orthogonal to each other (i.e. $\Delta W^0 \angle \Delta W^0_{BP} < 90°$), then it suggests that the two algorithms are specifying similar weight updates.

As in previous work (*Lillicrap et al., 2016*), we found that the initial weight updates for our network were orthogonal to the updates specified by backpropagation. But, as the network learned the angle dropped to approximately $65°$, before rising again slightly to roughly $70°$ (*Figure 7A*, blue line). This suggests that our network was learning to develop local weight updates in the hidden layer that were in rough agreement with the updates that explicit backpropagation would produce. However, this drop in orthogonality was still much less than that observed in non-spiking artificial neural networks learning with random feedback weights, which show a drop to below $45°$ (*Lillicrap et al., 2016*). We suspected that the higher angle between the weight updates that we observed may have been because we were using spikes to communicate the feedback from the

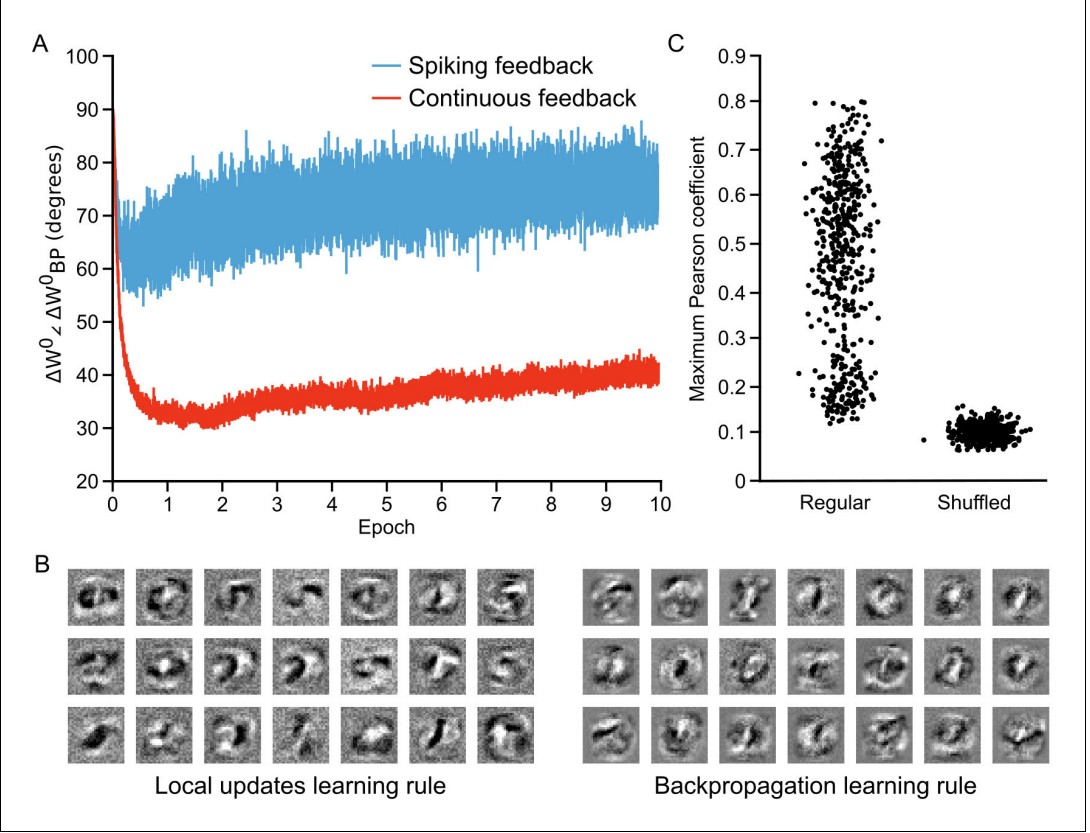

**Figure 7.** Approximation of backpropagation with local learning rules. (**A**) Plot of the angle between weight updates prescribed by our local update learning algorithm compared to those prescribed by backpropagation of error, for a one hidden layer network over 10 epochs of training (each point on the horizontal axis corresponds to one image presentation). Data was time-averaged using a sliding window of 100 image presentations. When training the network using the local update learning algorithm, feedback was sent to the hidden layer either using spiking activity from the output layer units (blue) or by directly sending the spike rates of output units (red). The angle between the local update $\Delta W^0$ and backpropagation weight updates $\Delta W_{BP}^0$ remains under 90° during training, indicating that both algorithms point weight updates in a similar direction. (**B**) Examples of hidden layer receptive fields (synaptic weights) obtained by training the network in A using our local update learning rule (left) and backpropagation of error (right) for 60 epochs. (**C**) Plot of correlation between local update receptive fields and backpropagation receptive fields. For each of the receptive fields produced by local update, we plot the maximum Pearson correlation coefficient between it and all 500 receptive fields learned using backpropagation (Regular). Overall, the maximum correlation coefficients are greater than those obtained after shuffling all of the values of the local update receptive fields (Shuffled).

DOI: https://doi.org/10.7554/eLife.22901.013

The following source data is available for figure 7:

**Source data 1.** Fig_7A.csv.

DOI: https://doi.org/10.7554/eLife.22901.014

upper layer, which could introduce both noise and bias in the estimates of the output layer activity. To test this, we also examined the weight updates that our algorithm would produce if we propagated the spike rates of the output layer neurons, $\boldsymbol{\phi}^1(t)$, back directly through the random feedback weights, $Y$. In this scenario, we observed a much sharper drop in the $\Delta W^0 \angle \Delta W_{BP}^0$ angle, which reduced to roughly 35° before rising again to 40° (**Figure 7A**, red line). These results show that, in principle, our algorithm is learning to approximate the backpropagation algorithm, though with some drop in accuracy introduced by the use of spikes to propagate output layer activities to the hidden layer.

To further examine how our local learning algorithm compared to backpropagation we compared the low-level features that the two algorithms learned. To do this, we trained the one hidden layer network with both our algorithm and backpropagation. We then examined the receptive fields (i.e. the synaptic weights) produced by both algorithms in the hidden layer synapses ($W^0$) after 60 epochs of training. The two algorithms produced qualitatively similar receptive fields (*Figure 7B*). Both produced receptive fields with clear, high-contrast features for detecting particular strokes or shapes. To quantify the similarity, we conducted pair-wise correlation calculations for the receptive fields produced by the two algorithms and identified the maximum correlation pairs for each. Compared to shuffled versions of the receptive fields, there was a very high level of maximum correlation (*Figure 7C*), showing that the receptive fields were indeed quite similar. Thus, the data demonstrate that our learning algorithm using random feedback weights into segregated dendrites can in fact come to approximate the backpropagation of error algorithm.

## Conditions on feedback weights

Once we had convinced ourselves that our learning algorithm was, in fact, providing a solution to the credit assignment problem, we wanted to examine some of the constraints on learning. First, we wanted to explore the structure of the feedback weights. In our initial simulations we used non-sparse, random (i.e. normally distributed) feedback weights. We were interested in whether learning could still work with sparse weights, given that neocortical connectivity is sparse. As well, we wondered whether symmetric weights would *improve* learning, which would be expected given previous findings (*Lillicrap et al., 2016*; *Lee et al., 2015*; *Liao et al., 2015*). To explore these questions, we trained our one hidden layer network using both sparse feedback weights (only 20% non-zero values) and symmetric weights ($Y = W^{1^T}$) (*Figure 8A,C*). We found that learning actually *improved* slightly with sparse weights (*Figure 8B*, red line), achieving an average error rate of 3.7% by the 60th epoch, compared to the average 4.1% error rate achieved with fully random weights. But, this result appeared to depend on the magnitude of the sparse weights. To compensate for the loss of 80% of the weights we initially increased the sparse synaptic weight magnitudes by a factor of 5. However, when we did not re-scale the sparse weights learning was actually *worse* (*Figure 8—figure supplement 1*), though this could likely be dealt with by a careful resetting of learning rates. Altogether, our results suggest that sparse feedback provides a signal that is sufficient for credit assignment.

Similar to sparse feedback weights, symmetric feedback weights also improved learning, leading to a rapid decrease in the test error and an error rate of 3.6% by the 60th epoch (*Figure 8D*, red line). This is interesting, given that backpropagation assumes symmetric feedback weights (*Lillicrap et al., 2016*; *Bengio et al., 2015*), though our proof of Theorem 1 does not. However, when we added noise to the symmetric weights any advantage was eliminated and learning was, in fact, slightly impaired (*Figure 8D*, blue line). At first, this was a very surprising result: given that learning works with random feedback weights, why would it not work with symmetric weights with noise? However, when we considered our previous finding that during the first epoch the feedforward weights, $W^1$, learn to have the feedforward Jacobian match the inverse of the feedback Jacobian (*Figure 5—figure supplement 1*) a possible answer emerges. In the case of symmetric feedback weights the synaptic matrix $Y$ is changing as $W^1$ changes. This works fine when $Y$ is set to $W^{1^T}$, since that artificially forces something akin to backpropagation. But, if the feedback weights are set to $W^{1^T}$ plus noise, then the system can never align the Jacobians appropriately, since $Y$ is now a moving target. This would imply that any implementation of feedback learning must either be very effective (to achieve the right feedback) or very slow (to allow the feedforward weights to adapt).

## Learning with partial apical attenuation

Another constraint that we wished to examine was whether total segregation of the apical inputs was necessary, given that real pyramidal neurons only show an attenuation of distal apical inputs to the soma (*Larkum et al., 1999*). Total segregation ($g_a = 0$) renders the network effectively feed-forward in its dynamics, which made it easier to construct the loss functions to ensure that reducing $L^0$ also reduces $L^1$ (see *Figure 5* and Theorem 1). But, we wondered whether some degree of apical conductance to the soma would be sufficiently innocuous so as to not disrupt deep learning. To examine this, we re-ran our two hidden layer network, but now, we allowed the apical dendritic

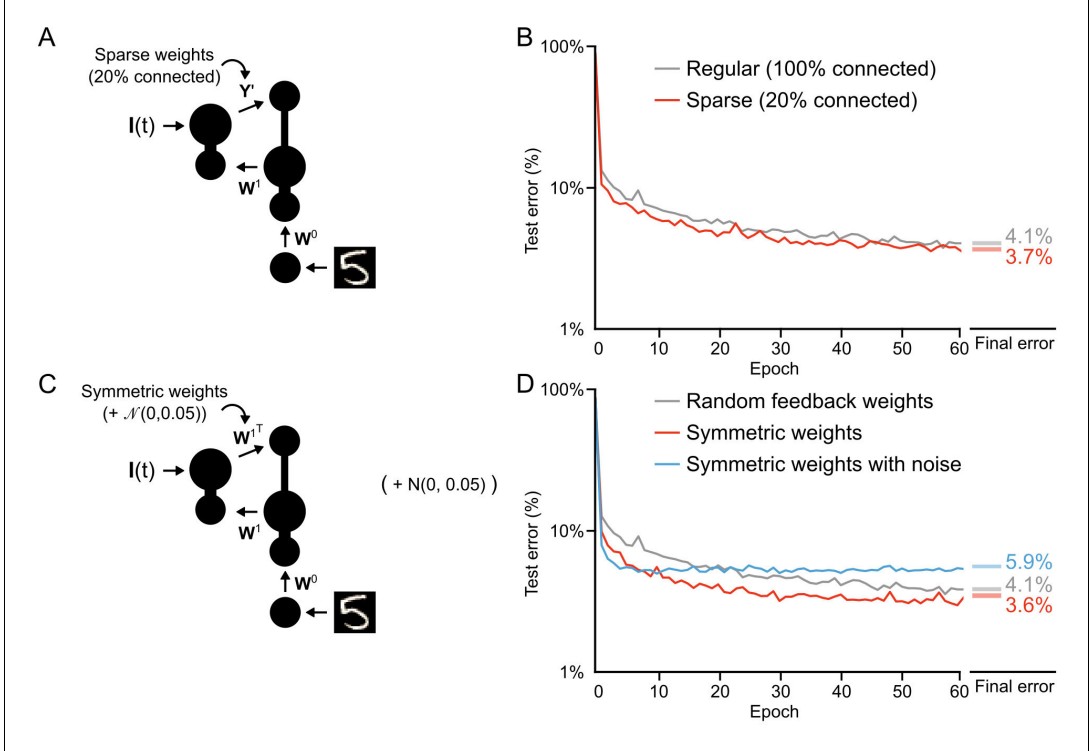

**Figure 8.** Conditions on feedback synapses for effective learning. (**A**) Diagram of a one hidden layer network trained in B, with 80% of feedback weights set to zero. The remaining feedback weights $Y'$ were multiplied by five in order to maintain a similar overall magnitude of feedback signals. (**B**) Plot of test error across 60 epochs for our standard one hidden layer network (gray) and a network with sparse feedback weights (red). Sparse feedback weights resulted in improved learning performance compared to fully connected feedback weights. *Right*: Spreads (min – max) of the results of repeated weight tests ($n = 20$) after 60 epochs for each of the networks. Percentages indicate mean final test errors for each network (two-tailed t-test, regular vs. sparse: $t_{38} = 16.43$, $p = 7.4 \times 10^{-19}$). (**C**) Diagram of a one hidden layer network trained in D, with feedback weights that are symmetric to feedforward weights $W^1$, and symmetric but with added noise. Noise added to feedback weights is drawn from a normal distribution with variance $\sigma = 0.05$. (**D**) Plot of test error across 60 epochs of our standard one hidden layer network (gray), a network with symmetric weights (red), and a network with symmetric weights with added noise (blue). Symmetric weights result in improved learning performance compared to random feedback weights, but adding noise to symmetric weights results in impaired learning. *Right*: Spreads (min – max) of the results of repeated weight tests ($n = 20$) after 60 epochs for each of the networks. Percentages indicate means (two-tailed t-test, random vs. symmetric: $t_{38} = 18.46$, $p = 4.3 \times 10^{-20}$; random vs. symmetric with noise: $t_{38} = -71.54$, $p = 1.2 \times 10^{-41}$; symmetric vs. symmetric with noise: $t_{38} = -80.35$, $p = 1.5 \times 10^{-43}$, Bonferroni correction for multiple comparisons).

DOI: https://doi.org/10.7554/eLife.22901.015

The following source data and figure supplement are available for figure 8:

**Source data 1.** Fig_8B_errors.csv.

DOI: https://doi.org/10.7554/eLife.22901.017

**Figure supplement 1.** Importance of weight magnitudes for learning with sparse weights.

DOI: https://doi.org/10.7554/eLife.22901.016

voltage to influence the somatic voltage by setting $g_a = 0.05$. This value gave us twelve times more attenuation than the attenuation from the basal compartments, since $g_b = 0.6$ (**Figure 9A**). When we compared the learning in this scenario to the scenario with total apical segregation, we observed very little difference in the error rates on the test set (**Figure 9B**, gray and red lines). Importantly, though, we found that if we increased the apical conductance to the same level as the basal ($g_a = g_b = 0.6$) then the learning was significantly impaired (**Figure 9B**, blue line). This demonstrates that although total apical attenuation is not necessary, partial segregation of the apical compartment from the soma is necessary. That result makes sense given that our local targets for the hidden layer neurons incorporate a term that is supposed to reflect the response of the output neurons to the feedforward sensory information ($\alpha^f$). Without some sort of separation of feedforward and feedback information, as is assumed in other models of deep learning (**Lillicrap et al., 2016**; **Lee et al., 2015**),

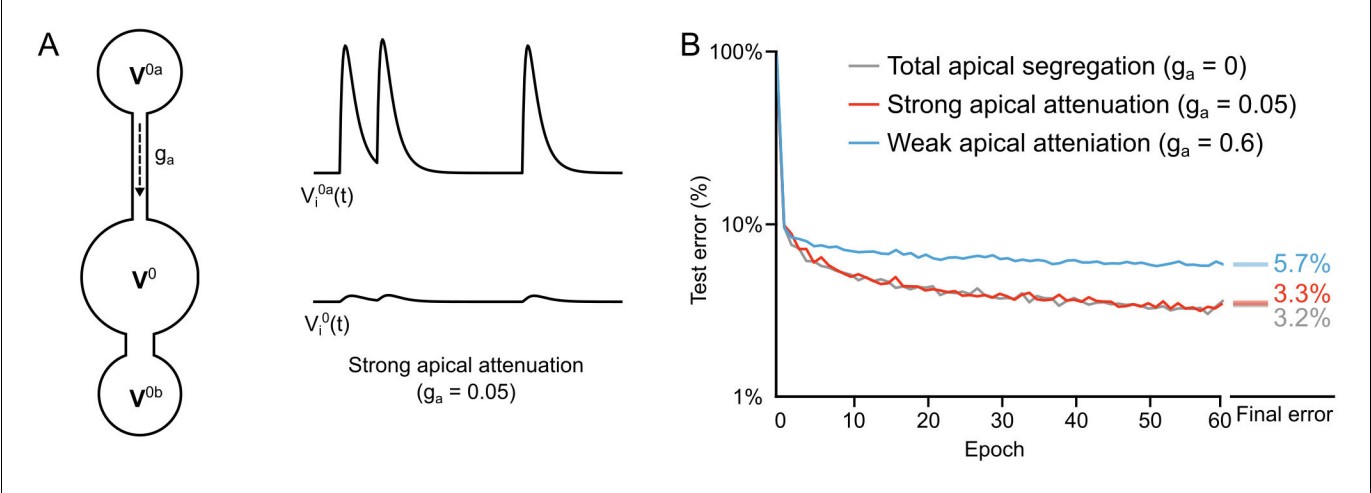

**Figure 9.** Importance of dendritic segregation for deep learning. (**A**) *Left*: Diagram of a hidden layer neuron. $g_a$ represents the strength of the coupling between the apical dendrite and soma. *Right*: Example traces of the apical voltage in a single neuron $V_i^{0a}$ and the somatic voltage $V_i^0$ in response to spikes arriving at apical synapses. Here $g_a = 0.05$, so the apical activity is strongly attenuated at the soma. (**B**) Plot of test error across 60 epochs of training on MNIST of a two hidden layer network, with total apical segregation (gray), strong apical attenuation (red) and weak apical attenuation (blue). Apical input to the soma did not prevent learning if it was strongly attenuated, but weak apical attenuation impaired deep learning. *Right*: Spreads (min – max) of the results of repeated weight tests ($n = 20$) after 60 epochs for each of the networks. Percentages indicate means (two-tailed t-test, total segregation vs. strong attenuation: $t_{38} = -4.00$, $p = 8.4 \times 10^{-4}$; total segregation vs. weak attenuation: $t_{38} = -95.24$, $p = 2.4 \times 10^{-46}$; strong attenuation vs. weak attenuation: $t_{38} = -92.51$, $p = 7.1 \times 10^{-46}$, Bonferroni correction for multiple comparisons).
DOI: https://doi.org/10.7554/eLife.22901.018

The following source data is available for figure 9:

**Source data 1.** Fig_9B_errors.csv.
DOI: https://doi.org/10.7554/eLife.22901.019

this feedback signal would get corrupted by recurrent dynamics in the network. Our data show that electrontonically segregated dendrites is one potential way to achieve the separation between feed-forward and feedback information that is required for deep learning.

## Discussion

Deep learning has radically altered the field of AI, demonstrating that parallel distributed processing across multiple layers can produce human/animal-level capabilities in image classification, pattern recognition and reinforcement learning (*Hinton et al., 2006*; *LeCun et al., 2015*; *Mnih et al., 2015*; *Silver et al., 2016*; *Krizhevsky et al., 2012*; *He et al., 2015*). Deep learning was motivated by analogies to the real brain (*LeCun et al., 2015*; *Cox and Dean, 2014*), so it is tantalizing that recent studies have shown that deep neural networks develop representations that strongly resemble the representations observed in the mammalian neocortex (*Khaligh-Razavi and Kriegeskorte, 2014*; *Yamins and DiCarlo, 2016*; *Cadieu et al., 2014*; *Kubilius et al., 2016*). In fact, deep learning models can match cortical representations better than some models that explicitly attempt to mimic the real brain (*Khaligh-Razavi and Kriegeskorte, 2014*). Hence, at a phenomenological level, it appears that deep learning, defined as multilayer cost function reduction with appropriate credit assignment, may be key to the remarkable computational prowess of the mammalian brain (*Marblestone et al., 2016*). However, the lack of biologically feasible mechanisms for credit assignment in deep learning algorithms, most notably backpropagation of error (*Rumelhart et al., 1986*), has left neuroscientists with a mystery. Given that the brain cannot use backpropagation, how does it solve the credit assignment problem (*Figure 1*)? Here, we expanded on an idea that previous authors have explored (*Körding and König, 2001*; *Spratling, 2002*; *Spratling and Johnson, 2006*) and demonstrated that segregating the feedback and feedforward inputs to neurons, much as the real neocortex does (*Larkum et al., 1999*; *2007*; *2009*), can enable the construction of local targets to assign credit appropriately to hidden layer neurons (*Figure 2*). With this formulation, we showed that we could

use segregated dendritic compartments to coordinate learning across layers (*Figure 3*, *Figure 4* and *Figure 5*). This enabled our network to take advantage of multiple layers to develop representations of hand-written digits in hidden layers that enabled better levels of classification accuracy on the MNIST dataset than could be achieved with a single layer (*Figure 6*). Furthermore, we found that our algorithm actually approximated the weight updates that would be prescribed by backpropagation, and produced similar low-level feature detectors (*Figure 7*). As well, we showed that our basic framework works with sparse feedback connections (*Figure 8*) and more realistic, partial apical attenuation (*Figure 9*). Therefore, our work demonstrates that deep learning is possible in a biologically feasible framework, provided that feedforward and feedback signals are sufficiently segregated in different dendrites.

In this work we adopted a similar strategy to the one taken by *Lee et al., 2015* in their difference target propagation algorithm, wherein the feedback from higher layers is used to construct local firing-rate targets at the hidden layers. One of the reasons that we adopted this strategy is that it is appealing to think that feedback from upper layers may not simply be providing a signal for plasticity, but also a predictive and/or modulatory signal to push the hidden layer neurons towards a 'better' activity pattern in *real-time*. This sort of top-down control could be used by the brain to improve sensory processing in different contexts and engage in inference (*Bengio et al., 2015*). Indeed, framing cortico-cortical feedback as a mechanism to predict or modulate incoming sensory activity is a more common way of viewing feedback signals in the neocortex (*Larkum, 2013*; *Gilbert and Li, 2013*; *Zhang et al., 2014*; *Fiser et al., 2016*; *Leinweber et al., 2017*). In light of this, it is interesting to note that distal apical inputs in sensory cortical areas can predict upcoming stimuli (*Leinweber et al., 2017*; *Fiser et al., 2016*), and help animals perform sensory discrimination tasks (*Takahashi et al., 2016*; *Manita et al., 2015*). However, in our model, we did not actually implement a system that altered the hidden layer activity to make sensory computations—we simply used the feedback signals to drive learning. In-line with this view of top-down feedback, two recent papers have found evidence that cortical feedback can indeed guide feedforward sensory plasticity (*Thompson et al., 2016*; *Yamada et al., 2017*), and in the hippocampus, there is evidence that plateau potentials generated by apical inputs are key determinants of plasticity (*Bittner et al., 2015*; *Bittner et al., 2017*). But, ultimately, there is no reason that feedback signals cannot provide both top-down predicton/modulation and a signal for learning (*Spratling, 2002*). In this respect, a potential future advance on our model would be to implement a system wherein the feedback makes predictions and 'nudges' the hidden layers towards appropriate activity patterns in order to guide learning and shape perception simultaneously. This proposal is reminiscent of the approach taken in previous computational models (*Urbanczik and Senn, 2014*; *Spratling and Johnson, 2006*; *Körding and König, 2001*). Future research could study how top-down control of activity and a signal for credit assignment can be combined.

In a number of ways, the model that we presented here is more biologically feasible than other deep learning models. We utilized leaky integrator neurons that communicate with spikes, we simulated in near continuous-time, and we used spatially local synaptic plasticity rules. Yet, there are still clearly unresolved issues of biological feasibility in our model. Most notably, the model updates synaptic weights using the difference between two plateau potentials that occur following two different phases. There are three issues with this method from a biological standpoint. First, it necessitates two distinct global phases of processing (the 'forward' and 'target' phases). Second, the plateau potentials occur in the apical compartment, but they are used to update the basal synapses, meaning that this information from the apical dendrites must somehow be communicated to the rest of the neuron. Third, the two plateau potentials occur with a temporal gap of tens of milliseconds, meaning that this difference must somehow be computed over time.

These issues could, theoretically, be resolved in a biologically realistic manner. The two different phases could be a result of a global signal indicating whether the teaching signal was present. This could be accomplished with neuromodulatory systems (*Pi et al., 2013*), or alternatively, with oscillations that the teaching signal and apical dendrites are phase locked to (*Veit et al., 2017*). Communicating plateau potentials to the basal dendrites is also possible using known biological principles. Plateau potentials induce bursts of action potentials in pyramidal neurons (*Larkum et al., 1999*), and the rate-of-fire of the bursts would be a function of the level of the plateau potential. Given that action potentials would propagate back through the basal dendrites (*Kampa and Stuart, 2006*), any cellular mechanism in the basal dendrites that is sensitive to rate-of-fire of bursts could be used to

detect the level of the plateau potentials in the apical dendrite. Finally, taking the difference between two events that occur tens of milliseconds apart is possible if such a hypothetical cellular signal that is sensitive to bursts had a slow decay time constant, and reacted differently depending on whether the global phase signal was active. A simple mathematical formulation for such a cellular signal is given in the methods (see *Equations (36) and (37)*). It is worth noting that incorporation of bursting into somatic dynamics would be unlikely to affect the learning results we presented here. This is because we calculate weight updates by averaging the activity of the neurons for a period after the network is near steady-state (i.e. the period marked with the blue line in *Figure 3C*, see also *Equation (5)*). Even if bursts of activity temporarily altered the dynamics of the network, they would not significantly alter the steady-state activity. Future work could expand on the model presented here and explore whether bursting activity might beneficially alter somatic dynamics (e.g. for on-line inference), as well as driving learning.

These possible implementations are clearly speculative, and only partially in-line with experimental evidence. As the adage goes, all models are wrong, but some models are useful. Our model aims to inspire new ways to think about how the credit assignment problem could be solved by known circuits in the brain. Our study demonstrates that some of the machinery that is known to exist in the neocortex, namely electrotonically segregated apical dendrites receiving top-down inputs, may be well-suited to credit assignment computations. What we are proposing is that the neocortex could use the segregation of top-down inputs to the apical dendrites in order to solve the credit assignment problem, without using a separate feedback pathway as is implicit in most deep learning models used in machine learning. We consider this to be the core insight of our model, and an important step in making deep learning more biologically plausible. Indeed, our model makes both a generic, and a specific, prediction about the role of synaptic inputs to apical dendrites during learning. *The generic prediction is that the sign of synaptic plasticity, that is whether LTP or LTD occur, in the basal dendrites will be modulated by different patterns of inputs to the apical dendrites*. The more specific prediction that our model makes is that the timing of apical inputs relative to basal inputs should be what determines the sign of plasticity for synapses in the basal dendrites. For example, if apical and basal inputs arrive at the same time, but the apical inputs disappear before the basal inputs do, then presumably plateau potentials will be stronger early in the stimulus presentation (i.e. $\alpha^f > \alpha^t$), and so the basal synapses should engage in LTD. In contrast, if the apical inputs only arrive after the basal inputs have been active for some period of time, then plateau potentials will be stronger towards the end of stimulus presentation (i.e. $\alpha^f < \alpha^t$), and so the basal synapses should engage in LTP. Both the generic and specific predictions should be experimentally testable using modern optical techniques to separate the inputs to the basal and apical dendrites (*Figure 10*).

Another direction for future research should be to consider how to use the machinery of neocortical microcircuits to communicate credit assignment signals without relying on differences across phases, as we did here. For example, somatostatin positive interneurons, which possess short-term facilitating synapses (*Silberberg and Markram, 2007*), are particularly sensitive to bursts of spikes, and could be part of a mechanism to calculate differences in the top-down signals being received by pyramidal neuron dendrites. If a calculation of this difference spanned the time before and after a teaching signal arrived, it could, theoretically, provide the computation that our system implements with a difference between plateau potentials. Indeed, we would argue that credit assignment may be one of the major functions of the canonical neocortical microcircuit motif. If this is correct, then the inhibitory interneurons that target apical dendrites may be used by the neocortex to control learning (*Murayama et al., 2009*). Although this is speculative, it is worth noting that current evidence supports the idea that neuromodulatory inputs carrying temporally precise salience information (*Hangya et al., 2015*) can shut off interneurons to disinhibit the distal apical dendrites (*Pi et al., 2013*; *Karnani et al., 2016*; *Pfeffer et al., 2013*; *Brombas et al., 2014*), and presumably, promote apical communication to the soma. Recent work suggests that the specific patterns of interneuron inhibition on the apical dendrites are spatially precise and differentially timed to motor behaviours (*Muñoz et al., 2017*), which suggests that there may well be coordinated physiological mechanisms for determining when and how cortico-cortical feedback is transmitted to the soma and basal dendrites. Future research should examine whether these inhibitory and neuromodulatory mechanisms do, in fact, control plasticity in the basal dendrites of pyramidal neurons, as our model, and some recent experimental work (*Bittner et al., 2015*; *Bittner et al., 2017*), would predict.

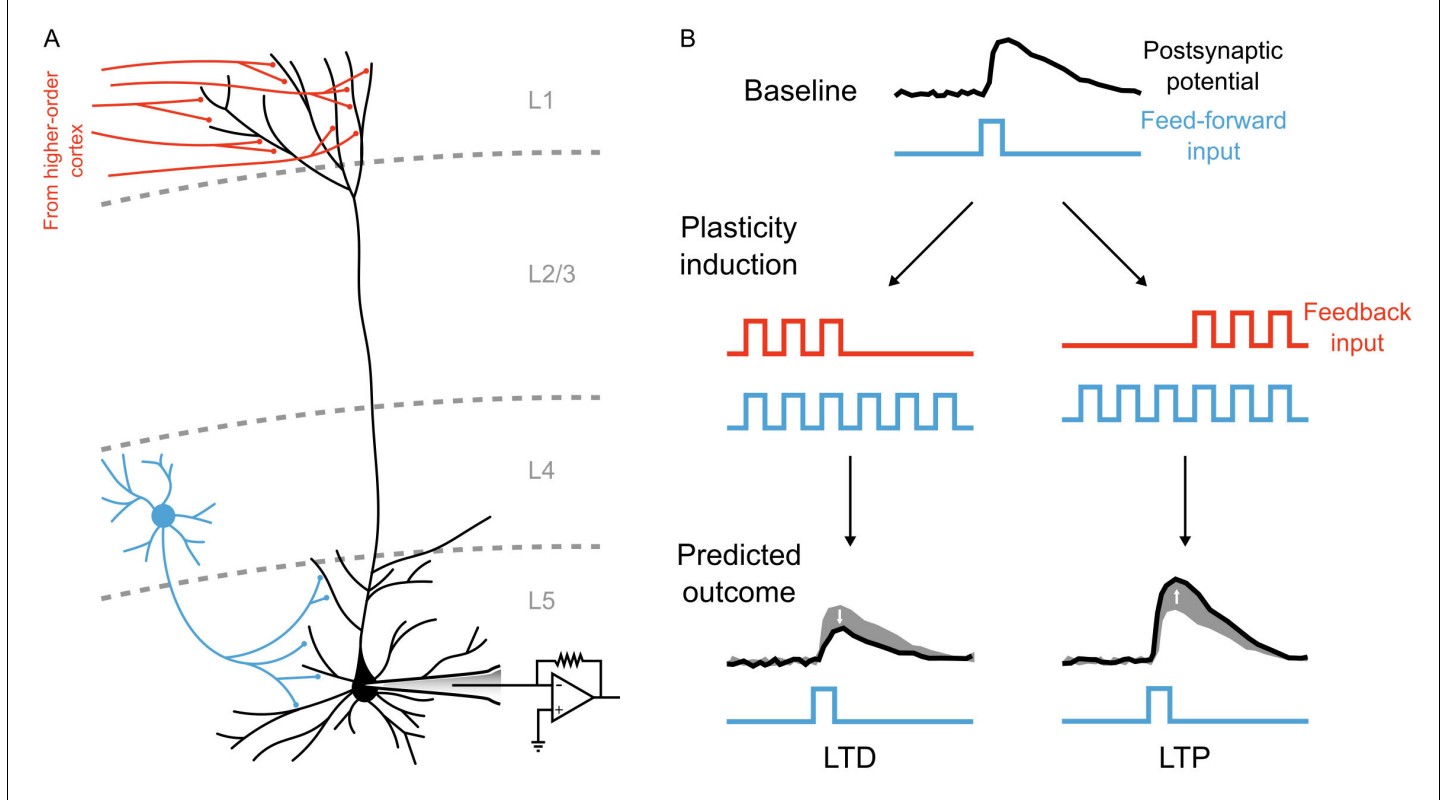

**Figure 10.** An experiment to test the central prediction of the model. (**A**) Illustration of the basic experimental set-up required to test the predictions (generic or specific) of the deep learning with segregated dendrites model. To test the predictions of the model, patch clamp recordings could be performed in neocortical pyramidal neurons (e.g. layer 5 neurons, shown in black), while the top-down inputs to the apical dendrites and bottom-up inputs to the basal dendrites are controlled separately. This could be accomplished optically, for example by infecting layer 4 cells with channelrhodopsin (blue cell), and a higher-order cortical region with a red-shifted opsin (red axon projections), such that the two inputs could be controlled by different colors of light. (**B**) Illustration of the specific experimental prediction of the model. With separate control of top-down and bottom-up inputs a synaptic plasticity experiment could be conducted to test the central prediction of the model, that is that the timing of apical inputs relative to basal inputs should determine the sign of plasticity at basal dendrites. After recording baseline postsynaptic responses (black lines) to the basal inputs (blue lines) a plasticity induction protocol could either have the apical inputs (red lines) arrive early during basal inputs (left) or late during basal inputs (right). The prediction of our model would be that the former would induce LTD in the basal synapses, while the later would induce LTP.
DOI: https://doi.org/10.7554/eLife.22901.020

A non-biological issue that should be recognized is that the error rates which our network achieved were by no means as low as can be achieved with artificial neural networks, nor at human levels of performance (*Lecun et al., 1998*; *Li et al., 2016*). As well, our algorithm was not able to take advantage of very deep structures (beyond two hidden layers, the error rate did not improve). In contrast, increasing the depth of networks trained with backpropagation can lead to performance improvements (*Li et al., 2016*). But, these observations do not mean that our network was not engaged in deep learning. First, it is interesting to note that although the backpropagation algorithm is several decades old (*Rumelhart et al., 1986*), it was long considered to be useless for training networks with more than one or two hidden layers (*Bengio and LeCun, 2007*). Indeed, it was only the use of layer-by-layer training that initially led to the realization that deeper networks can achieve excellent performance (*Hinton et al., 2006*). Since then, both the use of very large datasets (with millions of examples), and additional modifications to the backpropagation algorithm, have been key to making backpropagation work well on deeper networks (*Sutskever et al., 2013*; *LeCun et al., 2015*). Future studies could examine how our algorithm could incorporate current techniques used in machine learning to work better on deeper architectures. Second, we stress that our network was not designed to match the state-of-the-art in machine learning, nor human capabilities. To test our basic hypothesis (and to run our leaky-integration and spiking simulations in a

reasonable amount of time) we kept the network small, we stopped training before it reached its asymptote, and we did not implement any add-ons to the learning to improve the error rates, such as convolution and pooling layers, initialization tricks, mini-batch training, drop-out, momentum or RMSProp (*Sutskever et al., 2013*; *Tieleman and Hinton, 2012*; *Srivastava et al., 2014*). Indeed, it would be quite surprising if a relatively vanilla, small network like ours could come close to matching current performance benchmarks in machine learning. Third, although our network was able to take advantage of multiple layers to improve the error rate, there may be a variety of reasons that ever increasing depth didn't improve performance significantly. For example, our use of direct connections from the output layer to the hidden layers may have impaired the network's ability to coordinate synaptic updates between *hidden* layers. As well, given our finding that the use of spikes produced weight updates that were less well-aligned to backpropagation (*Figure 7A*) it is possible that deeper architectures require mechanisms to overcome the inherent noisiness of spikes.

One aspect of our model that we did not develop was the potential for learning at the feedback synapses. Although we used random synaptic weights for feedback, we also demonstrated that our model actually learns to meet the mathematical conditions required for credit assignment (*Figure 5—figure supplement 1*). This suggests that it would be beneficial to develop a synaptic weight update rule for the feedback synapses that made this aspect of the learning better. Indeed, *Lee et al., 2015* implemented an 'inverse loss function' for their feedback synapses which promoted the development of feedforward and feedback functions that were roughly inverses of each other, leading to the emergence of auto-encoder functions in their network. In light of this, it is interesting to note that there is evidence for unique, 'reverse' spike-timing-dependent synaptic plasticity rules in the distal apical dendrites of pyramidal neurons (*Sjöström and Häusser, 2006*; *Letzkus et al., 2006*), which have been shown to produce symmetric feedback weights and auto-encoder functions in artificial spiking networks (*Burbank and Kreiman, 2012*; *Burbank, 2015*). Thus, it is possible that early in development the neocortex actually learns cortico-cortical feedback connections that help it to assign credit for later learning. Our work suggests that any experimental evidence showing that feedback connections learn to approximate the inverse of feedforward connections could be considered as evidence for deep learning in the neocortex.

A final consideration, which is related to learning at feedback synapses, is the likely importance of unsupervised learning for the real brain, that is learning without a teaching signal. In this paper, we focused on a supervised learning task with a teaching signal. Supervised learning certainly could occur in the brain, especially for goal-directed sensorimotor tasks where animals have access to examples that they could use to generate internal teaching signals *Teşileanu et al. (2017)*. But, unsupervised learning is likely critical for understanding the development of cognition (*Marblestone et al., 2016*). Importantly, unsupervised learning in multilayer networks still requires a solution to the credit assignment problem (*Bengio et al., 2015*), so our work here is not completely inapplicable. Nonetheless, future research should examine how the credit assignment problem can be addressed in the specific case of unsupervised learning.

In summary, deep learning has had a huge impact on AI, but, to date, its impact on neuroscience has been limited. Nonetheless, given a number of findings in neurophysiology and modeling (*Yamins and DiCarlo, 2016*), there is growing interest in understanding how deep learning may actually be achieved by the real brain (*Marblestone et al., 2016*). Our results show that by moving away from point neurons, and shifting towards multi-compartment neurons that segregate feedforward and feedback signals, the credit assignment problem can be solved and deep learning can be achieved. Perhaps the dendritic anatomy of neocortical pyramidal neurons is important for nature's own deep learning algorithm.

## Materials and methods

Code for the model can be obtained from a GitHub repository (https://github.com/jordan-g/Segregated-Dendrite-Deep-Learning) (*Guerguiev, 2017*), with a copy archived at https://github.com/elifesciences-publications/Segregated-Dendrite-Deep-Learning. For notational simplicity, we describe our model in the case of a network with only one hidden layer. We describe how this is extended to a network with multiple layers at the end of this section. As well, at the end of this section in *Table 1* we provide a table listing the parameter values we used for all of the simulations presented in this paper.

**Table 1.** List of parameter values used in our simulations.

| Parameter | Units | Value | Description |
|---|---|---|---|
| $dt$ | ms | 1 | Time step resolution |
| $\phi_{\max}$ | Hz | 200 | Maximum spike rate |
| $\tau_s$ | ms | 3 | Short synaptic time constant |
| $\tau_L$ | ms | 10 | Long synaptic time constant |
| $\Delta t_s$ | ms | 30 | Settle duration for calculation of average voltages |
| $g_b$ | S | 0.6 | Hidden layer conductance from basal dendrites to the soma |
| $g_a$ | S | 0, 0.05, 0.6 | Hidden layer conductance from apical dendrites to the soma |
| $g_d$ | S | 0.6 | Output layer conductance from dendrites to the soma |
| $g_l$ | S | 0.1 | Leak conductance |
| $V^R$ | mV | 0 | Resting membrane potential |
| $C_m$ | F | 1 | Membrane capacitance |
| $P_0$ | – | $20/\phi_{\max}$ | Hidden layer error signal scaling factor |
| $P_1$ | – | $20/\phi_{\max}^2$ | Output layer error signal scaling factor |

DOI: https://doi.org/10.7554/eLife.22901.021

## Neuronal dynamics

The network described here consists of an input layer with $\ell$ neurons, a hidden layer with $m$ neurons, and an output layer with $n$ neurons. Neurons in the input layer are simple Poisson spiking neurons whose rate-of-fire is determined by the intensity of image pixels (ranging from 0 - $\phi_{\max}$). Neurons in the hidden layer are modeled using three functional compartments—basal dendrites with voltages $\boldsymbol{V}^{0b}(t) = [V_1^{0b}(t), V_2^{0b}(t), ..., V_m^{0b}(t)]$, apical dendrites with voltages $\boldsymbol{V}^{0a}(t) = [V_1^{0a}(t), V_2^{0a}(t), ..., V_m^{0a}(t)]$, and somata with voltages $\boldsymbol{V}^0(t) = [V_1^0(t), V_2^0(t), ..., V_m^0(t)]$. Feedforward inputs from the input layer and feedback inputs from the output layer arrive at basal and apical synapses, respectively. At basal synapses, presynaptic spikes from input layer neurons are translated into filtered spike trains $\boldsymbol{s}^{\text{input}}(t) = [s_1^{\text{input}}(t), s_2^{\text{input}}(t), ..., s_\ell^{\text{input}}(t)]$ given by:

$$s_j^{\text{input}}(t) = \sum_k \kappa(t - t_{jk}^{\text{input}}) \tag{11}$$

where $t_{jk}^{\text{input}}$ is the $k$ th spike time of input neuron $j$ is the response kernel given by:

$$\kappa(t) = (e^{-t/\tau_L} - e^{-t/\tau_s})\Theta(t)/(\tau_L - \tau_s)$$

where $\tau_s$ and $\tau_L$ are short and long time constants, and $\Theta$ is the Heaviside step function. Since the network is fully-connected, each neuron in the hidden layer will receive the same set of filtered spike trains from input layer neurons. The filtered spike trains at apical synapses, $\boldsymbol{s}^1(t) = [s_1^1(t), s_2^1(t), ..., s_n^1(t)]$, are modeled in the same manner. The basal and apical dendritic potentials for neuron $i$ are then given by weighted sums of the filtered spike trains at either its basal or apical synapses:

$$
\begin{aligned}
V_i^{0b}(t) &= \sum_{j=1}^{\ell} W_{ij}^0 s_j^{\text{input}}(t) + b_i^0 \\
V_i^{0a}(t) &= \sum_{j=1}^{n} Y_{ij} s_j^1(t)
\end{aligned}
\tag{13}
$$

where $\boldsymbol{b}^0 = [b_1^0, b_2^0, ..., b_m^0]$ are bias terms, $\boldsymbol{W}^0$ is the $m \times \ell$ matrix of feedforward weights for neurons in the hidden layer, and $\boldsymbol{Y}$ is the $m \times n$ matrix of their feedback weights. The somatic voltage for neuron $i$ evolves with leak as:

$$\tau \frac{dV_i^0(t)}{dt} = (V^R - V_i^0(t)) + \frac{g_b}{g_l}(V_i^{0b}(t) - V_i^0(t)) + \frac{g_a}{g_l}(V_i^{0a}(t) - V_i^0(t)) \tag{14}$$

$$= (V^R - V_i^0(t)) + \frac{g_b}{g_l}\left(\sum_{j=1}^{\ell} W_{ij}^0 s_j^{\text{input}}(t) + b_i^0 - V_i^0(t)\right) + \frac{g_a}{g_l}\left(\sum_{j=1}^{n} Y_{ij}^0 s_j^1(t) - V_i^0(t)\right) \tag{15}$$

where $V^R$ is the resting potential, $g_l$ is the leak conductance, $g_b$ is the conductance from the basal dendrite to the soma, and $g_a$ is the conductance from the apical dendrite to the soma, and $\tau$ is a function of $g_l$ and the membrane capacitance $C_m$:

$$\tau = \frac{C_m}{g_l} \tag{16}$$

Note that for simplicity's sake we are assuming a resting potential of 0 mV and a membrane capacitance of 1 F, but these values are not important for the results. *Equations (13) and (14)* are identical to the *Equation (1)* in results.

The instantaneous firing rates of neurons in the hidden layer are given by $\boldsymbol{\phi}^0(t) = [\phi_1^0(t), \phi_2^0(t), ..., \phi_m^0(t)]$, where $\phi_i^0(t)$ is the result of applying a nonlinearity, $\sigma(\cdot)$, to the somatic potential $V_i^0(t)$. We chose $\sigma(\cdot)$ to be a simple sigmoidal function, such that:

$$\phi_i^0(t) = \phi_{\max}\sigma(V_i^0(t)) = \phi_{\max}\frac{1}{1 + e^{-V_i^0(t)}} \tag{17}$$

Here, $\phi_{\max}$ is the maximum possible rate-of-fire for the neurons, which we set to 200 Hz. Note that *Equation (17)* is identical to *Equation (3)* in results. Spikes are then generated using Poisson processes with these firing rates. We note that although the maximum rate was 200 Hz, the neurons rarely achieved anything close to this rate, and the average rate of fire in the neurons during our simulations was 24 Hz.

Units in the output layer are modeled using only two compartments, dendrites with voltages $\boldsymbol{V}^{1b}(t) = [V_1^{1b}(t), V_2^{1b}(t), ..., V_n^{1b}(t)]$ and somata with voltages $\boldsymbol{V}^1(t) = [V_1^1(t), V_2^1(t), ..., V_n^1(t)]$ is given by:

$$V_i^{1b}(t) = \sum_{j=1}^{m} W_{ij}^1 s_j^0(t) + b_i^1 \tag{18}$$

where $s^0(t) = [s_1^0(t), s_2^0(t), ..., s_m^0(t)]$ are the filtered presynaptic spike trains at synapses that receive feedforward input from the hidden layer, and are calculated in the manner described by *Equation (11)*. $V_i^1(t)$ evolves as:

$$\tau \frac{dV_i^1(t)}{dt} = (V^R - V_i^1(t)) + \frac{g_d}{g_l}(V_i^{1b}(t) - V_i^1(t)) + I_i(t) \tag{19}$$

where $g_l$ is the leak conductance, $g_d$ is the conductance from the dendrite to the soma, and $\boldsymbol{I}(t) = [I_1(t), I_2(t), ..., I_n(t)]$ are somatic currents that can drive output neurons toward a desired somatic voltage. For neuron $i$, $I_i$ is given by:

$$I_i(t) = g_{E_i}(t)(E_E - V_i^1(t)) + g_{I_i}(t)(E_I - V_i^1(t)) \tag{20}$$

where $\boldsymbol{g_E}(t) = [g_{E_1}(t), g_{E_2}(t), ..., g_{E_n}(t)]$ and $\boldsymbol{g_I}(t) = [g_{I_1}(t), g_{I_2}(t), ..., g_{I_n}(t)]$ are time-varying excitatory and inhibitory nudging conductances, and $E_E$ and $E_I$ are the excitatory and inhibitory reversal potentials. In our simulations, we set $E_E = 8$ V and $E_I = -8$ V. During the target phase only, we set $g_{I_i} = 1$ and $g_{E_i} = 0$ for all units $i$ whose output should be minimal, and $g_{E_i} = 1$ and $g_{I_i} = 0$ for the unit whose output should be maximal. In this way, all units other than the 'target' unit are silenced, while the 'target' unit receives a strong excitatory drive. In the forward phase, $\boldsymbol{I}(t)$ is set to 0. The Poisson spike rates $\boldsymbol{\phi}^1(t) = [\phi_1^1(t), \phi_2^1(t), ..., \phi_n^1(t)]$ are calculated as in *Equation (17)*.

## Plateau potentials

At the end of the forward and target phases, we calculate plateau potentials $\boldsymbol{\alpha}^f = [\alpha_1^f, \alpha_2^f, ..., \alpha_m^f]$ and $\boldsymbol{\alpha}^t = [\alpha_1^t, \alpha_2^t, ..., \alpha_m^t]$ for apical dendrites of hidden layer neurons, where $\alpha_i^f$ and $\alpha_i^t$ are given by:

$$\alpha_i^f = \sigma\left(\frac{1}{\Delta t_1}\int_{t_1-\Delta t_1}^{t_1} V_i^{0a}(t)dt\right)$$

$$\alpha_i^t = \sigma\left(\frac{1}{\Delta t_2}\int_{t_2-\Delta t_2}^{t_2} V_i^{0a}(t)dt\right)$$

(21)

where $t_1$ and $t_2$ are the end times of the forward and target phases, respectively, $\Delta t_s = 30$ ms is the settling time for the voltages, and $\Delta t_1$ and $\Delta t_2$ are given by:

$$\Delta t_1 = t_1 - (t_0 + \Delta t_s)$$
$$\Delta t_2 = t_2 - (t_1 + \Delta t_s)$$

(22)

Note that *Equation (21)* is identical to *Equation (5)* in results. These plateau potentials are used by hidden layer neurons to update their basal weights.

## Weight updates

All feedforward synaptic weights are updated at the end of each target phase. Output layer units update their synaptic weights $\boldsymbol{W}^1$ in order to minimize the loss function

$$L^1 = ||\boldsymbol{\phi}^{1*} - \phi_{\max}\sigma(\overline{\boldsymbol{V}^{1}}^f)||_2^2$$

(23)

where $\boldsymbol{\phi}^{1*} = \overline{\boldsymbol{\phi}^1}^t$ as in *Equation (6)*. Note that, as long as neuronal units calculate averages after the network has reached a steady state, and the firing-rates of the neurons are in the linear region of the sigmoid function, then for layer $x$,

$$\phi_{\max}\sigma(\overline{\boldsymbol{V}^{x}}^f) \approx \phi_{\max}\overline{\sigma(\boldsymbol{V}^x)}^f$$
$$= \overline{\boldsymbol{\phi}^{x}}^f$$

(24)

Thus,

$$L^1 \approx ||\overline{\boldsymbol{\phi}^1}^t - \overline{\boldsymbol{\phi}^{1}}^f||_2^2$$

(25)

as in *Equation (7)*.

All average voltages are calculated after a delay $\Delta t_s$ from the start of a phase, which allows for the network to reach a steady state before averaging begins. In practice this means that the average somatic voltage for output layer neuron $i$ in the forward phase, $\overline{V_i^{1}}^f$, has the property

$$\overline{V_i^{1}}^f \approx k_d\overline{V_i^{1b}}^f = k_d\left(\sum_{j=1}^{m} W_{ij}^1\overline{s_j^{0}}^f + b_i^1\right)$$

(26)

where $k_d$ is given by:

$$k_d = \frac{g_d}{g_l + g_d}$$

(27)

Thus,

$$\frac{\partial L^1}{\partial \boldsymbol{W}^1} \approx -k_d\phi_{\max}(\boldsymbol{\phi}^{1*} - \phi_{\max}\sigma(\overline{\boldsymbol{V}^{1}}^f))\sigma'(\overline{\boldsymbol{V}^{1}}^f)\circ\overline{\boldsymbol{s}^{0}}^f$$

$$\frac{\partial L^1}{\partial \boldsymbol{b}^1} \approx -k_d\phi_{\max}(\boldsymbol{\phi}^{1*} - \phi_{\max}\sigma(\overline{\boldsymbol{V}^{1}}^f))\sigma'(\overline{\boldsymbol{V}^{1}}^f)$$

(28)

Note that these partial derivatives assume that the activity during the target phase is *fixed*. We do this because the goal of learning is to have the network behave as it does during the target phase, even when the teaching signal is present. Thus, we do not update synapses in order to alter the target phase activity. As a result, there are no terms in the equation related to the partial derivatives of the voltages or firing-rates during the target phase.

The dendrites in the output layer use this approximation of the gradient in order to update their weights using gradient descent:

$$\boldsymbol{W}^1 \rightarrow \boldsymbol{W}^1 - \eta^1 P^1 \frac{\partial L^1}{\partial \boldsymbol{W}^1}$$
$$\boldsymbol{b}^1 \rightarrow \boldsymbol{b}^1 - \eta^1 P^1 \frac{\partial L^1}{\partial \boldsymbol{b}^1}$$

(29)

where $\eta^1$ is a learning rate constant, and $P^1$ is a scaling factor used to normalize the scale of the rate-of-fire function.

In the hidden layer, basal dendrites update their synaptic weights $\boldsymbol{W}^0$ by minimizing the loss function

$$L^0 = ||\boldsymbol{\phi}^{0*} - \phi_{\max}\sigma(\overline{\boldsymbol{V}^0}^f)||_2^2$$

(30)

We define the target rates-of-fire $\boldsymbol{\phi}^{0*} = [\phi_1^{0*}, \phi_2^{0*}, ..., \phi_m^{0*}]$ such that

$$\phi_i^{0*} = \overline{\phi_i^0}^f + \alpha_i^t - \alpha_i^f$$

(31)

where $\boldsymbol{\alpha}^f = [\alpha_1^f, \alpha_2^f, ..., \alpha_m^f]$ and $\boldsymbol{\alpha}^t = [\alpha_1^t, \alpha_2^t, ..., \alpha_m^t]$ are forward and target phase plateau potentials given in *Equation (21)*. Note that *Equation (31)* is identical to *Equation (8)* in results. These hidden layer target firing rates are similar to the targets used in difference target propagation (*Lee et al., 2015*).

Using *Equation (24)*, we can show that

$$L^0 \approx ||\boldsymbol{\alpha}^t - \boldsymbol{\alpha}^f||_2^2$$

(32)

as in *Equation (9)*. Hence:

$$\frac{\partial L^0}{\partial \boldsymbol{W}^0} \approx -k_b(\boldsymbol{\alpha}^t - \boldsymbol{\alpha}^f)\phi_{\max}\sigma'(\overline{\boldsymbol{V}^0}^f) \circ \overline{\boldsymbol{s}^{\text{input}}}^f$$
$$\frac{\partial L^0}{\partial \boldsymbol{b}^0} \approx -k_b(\boldsymbol{\alpha}^t - \boldsymbol{\alpha}^f)\phi_{\max}\sigma'(\overline{\boldsymbol{V}^0}^f)$$

(33)

where $k_b$ is given by:

$$k_b = \frac{g_b}{g_l + g_b + g_a}$$

(34)

Note that although $\boldsymbol{\phi}^{0*}$ is a function of $\boldsymbol{W}^0$ and $\boldsymbol{b}^0$, we do not differentiate this term with respect to the weights and biases. Instead, we treat $\boldsymbol{\phi}^{0*}$ as a fixed state for the hidden layer neurons to learn to reproduce. Basal weights are updated in order to descend this approximation of the gradient:

$$\boldsymbol{W}^0 \rightarrow \boldsymbol{W}^0 - \eta^0 P^0 \frac{\partial L^0}{\partial \boldsymbol{W}^0}$$
$$\boldsymbol{b}^0 \rightarrow \boldsymbol{b}^0 - \eta^0 P^0 \frac{\partial L^0}{\partial \boldsymbol{b}^0}$$

(35)

Again, we assume that the activity during the target phase is fixed, so no derivatives are taken with respect to voltages or firing-rates during the target phase.

Importantly, this update rule is spatially local for the hidden layer neurons. It consists essentially of three terms, (1) the difference in the plateau potentials for the target and forward phases ($\boldsymbol{\alpha}^t - \boldsymbol{\alpha}^f$), (2) the derivative of the spike rate function ($\phi_{\max}\sigma'(\overline{\boldsymbol{V}^0}^f)$), and (3) the filtered presynaptic spike trains ($\overline{\boldsymbol{s}^{\text{input}}}^f$). All three of these terms are values that a real neuron could theoretically calculate using some combination of molecular synaptic tags, calcium currents, and back-propagating action potentials.

One aspect of this update rule that is biologically questionable, though, is the use of the term ($\boldsymbol{\alpha}^t - \boldsymbol{\alpha}^f$). This requires a difference between plateau potentials that are separated by tens of

milliseconds. How could such a signal be used by basal dendrite synapses to guide their updates? Plateau potentials can drive bursts of spikes (*Larkum et al., 1999*), which can propagate to basal dendrites (*Kampa and Stuart, 2006*). Since the plateau potentials are similar to rate variables (i.e. a sigmoid applied to the voltage), the number of spikes during the bursts, $N^f = [N^f_1, N^f_2, ..., N^f_m]$ and $N^t = [N^t_1, N^t_2, ..., N^t_m]$, for the forward and target plateaus, respectively, could be sampled from a Poisson distribution with rate parameter equal to the plateau potential level:

$$N^f \sim Poisson(\alpha^f)$$
$$N^t \sim Poisson(\alpha^t)$$

(36)

If the distinct phases (forward and target) were marked by some global signal, $\phi(t)$, that was communicated to all of the neurons, for example a neuromodulatory signal, the phase of a global oscillation, or some blanket inhibition signal, then we can imagine an internal cellular memory mechanism in the basal dendrites of the $i^{th}$ neuron, $M_i$ (e.g. a molecular signal like the activity of an enzyme, the phosphorylation level of some protein, or the amount of calcium released from intracellular stores), which could be differentially sensitive to the inter-spike interval of bursts, depending on $\phi$. So, for example, if we define:

$$\phi(t) = \begin{cases} -1, & \text{if in the forward phase, i.e. } x = f \\ 1, & \text{if in the target phase, i.e. } x = t \end{cases}$$
$$\frac{dM_i(t)}{dt} \propto \phi(t)N^x_i$$

(37)

where x indicates the forward or target phase. Then, the change in $M_i$ from before the bursts occur to afterwards would be, on average, proportional to the difference $(\boldsymbol{\alpha}^t - \boldsymbol{\alpha}^f)$, and could be used to calculate the weight updates.

However, this is highly speculative, and it is not clear that such a mechanism would be present in real neurons. We have outlined the mathematics here to make the reality of implementing the current model explicit, but we would predict that the brain would have some alternative method for calculating differences between top-down inputs at different times, for example by using somatostatin positive interneurons that are preferentially sensitive to bursts and which target the apical dendrite (*Silberberg and Markram, 2007*). We are ultimately agnostic as to this mechanism, and so, it was not included in the current model.

## Multiple hidden layers

In order to extend our algorithm to deeper networks with multiple hidden layers, our model incorporates direct synaptic connections from the output layer to each hidden layer. Thus, each hidden layer receives feedback from the output layer through its own separate set of fixed, random weights. For example, in a network with two hidden layers, both layers receive the feedback from the output layer at their apical dendrites through backward weights $Y^0$ and $Y^1$. The local targets at each layer are then given by:

$$\phi^{2*} = \overline{\boldsymbol{\phi}^2}^t$$

(38)

$$\phi^{1*} = \overline{\phi^1}^t + \boldsymbol{\alpha}^{1t} - \boldsymbol{\alpha}^{1f}$$

(39)

$$\phi^{0*} = \overline{\phi^0}^t + \boldsymbol{\alpha}^{0t} - \boldsymbol{\alpha}^{0f}$$

(40)

where the superscripts $^0$ and $^1$ denote the first and second hidden layers, respectively, and the superscript $^2$ denotes the output layer.

The local loss functions at each layer are:

$$L^2 = ||\boldsymbol{\phi}^{2*} - \phi_{\max}\sigma(\overline{V^2}^f)||^2_2$$
$$L^1 = ||\boldsymbol{\phi}^{1*} - \phi_{\max}\sigma(\overline{V^1}^f)||^2_2$$
$$L^0 = ||\boldsymbol{\phi}^{0*} - \phi_{\max}\sigma(\overline{V^0}^f)||^2_2$$

(41)

where $L^2$ is the loss at the output layer. The learning rules used by the hidden layers in this scenario are the same as in the case with one hidden layer.

## Learning rate optimization

For each of the three network sizes that we present in this paper, a grid search was performed in order to find good learning rates. We set the learning rate for each layer by stepping through the range $[0.1, 0.3]$ with a step size of $0.02$. For each combination of learning rates, a neural network was trained for one epoch on the 60, 000 training examples, after which the network was tested on 10,000 test images. The learning rates that gave the best performance on the test set after an epoch of training were used as a basis for a second grid search around these learning rates that used a smaller step size of $0.01$. From this, the learning rates that gave the best test performance after 20 epochs were chosen as our learning rates for that network size.

In all of our simulations, we used a learning rate of 0.19 for a network with no hidden layers, learning rates of 0.21 (output and hidden) for a network with one hidden layer, and learning rates of 0.23 (hidden layers) and 0.12 (output layer) for a network with two hidden layers. All networks with one hidden layer had 500 hidden layer neurons, and all networks with two hidden layers had 500 neurons in the first hidden layer and 100 neurons in the second hidden layer.

## Training paradigm

For all simulations described in this paper, the neural networks were trained on classifying handwritten digits using the MNIST database of 28 pixel × 28 pixel images. Initial feedforward and feedback weights were chosen randomly from a uniform distribution over a range that was calculated to produce voltages in the dendrites between $-6$ - 12 V.

Prior to training, we tested a network's initial performance on a set of 10,000 test examples. This set of images was shuffled at the beginning of testing, and each example was shown to the network in sequence. Each input image was encoded into Poisson spiking activity of the 784 input neurons representing each pixel of the image. The firing rate of an input neuron was proportional to the brightness of the pixel that it represents (with spike rates between $[0 - \phi_{\max}]$. The spiking activity of each of the 784 input neurons was received by the neurons in the first hidden layer. For each test image, the network underwent only a forward phase. At the end of this phase, the network's classification of the input image was given by the neuron in the output layer with the greatest somatic potential (and therefore the greatest spike rate). The network's classification was compared to the target classification. After classifying all 10,000 testing examples, the network's classification error was given by the percentage of examples that it did not classify correctly.

Following the initial test, training of the neural network was done in an on-line fashion. All 60,000 training images were randomly shuffled at the start of each training epoch. The network was then shown each training image in sequence, undergoing a forward phase ending with a plateau potential, and a target phase ending with another plateau potential. All feedforward weights were then updated at the end of the target phase. At the end of the epoch (after all 60,000 images were shown to the network), the network was again tested on the 10,000 test examples. The network was trained for up to 60 epochs.

## Simulation details

For each training example, a minimum length of 50 ms was used for each of the forward and target phases. The lengths of the forward and target training phases were determined by adding their minimum length to an extra length term, which was chosen randomly from a Wald distribution with a mean of 2 ms and scale factor of 1. During testing, a fixed length of 500 ms was used for the forward transmit phase. Average forward and target phase voltages were calculated after a settle duration of $\Delta t_s = 30$ ms from the start of the phase.

For simulations with randomly sampled plateau potential times (*Figure 5—figure supplement 1*), the time at which each neuron's plateau potential occurred was randomly sampled from a folded normal distribution ($\mu = 0, \sigma^2 = 3$) that was truncated ($\max = 5$) such that plateau potentials occurred between 0 ms and 5 ms before the start of the next phase. In this scenario, the average apical voltage in the last 30 ms was averaged in the calculation of the plateau potential for a particular neuron.

The time-step used for simulations was $dt = 1$ ms. At each time-step, the network's state was updated bottom-to-top beginning with the first hidden layer and ending with the output layer. For each layer, dendritic potentials were updated, followed by somatic potentials, and finally their spiking activity. *Table 1* lists the simulation parameters and the values that were used in the figures presented.

All code was written using the Python programming language version 2.7 (RRID: SCR_008394) with the NumPy (RRID: SCR_008633) and SciPy (RRID: SCR_008058) libraries. The code is open source and is freely available at https://github.com/jordan-g/Segregated-Dendrite-Deep-Learning (*Guerguiev, 2017*). The data used to train the network was from the Mixed National Institute of Standards and Technology (MNIST) database, which is a modification of the original database from the National Institute of Standards and Technology (RRID: SCR_006440) (*Lecun et al., 1998*). The MNIST database can be found at http://yann.lecun.com/exdb/mnist/. Some of the simulations were run on the SciNet High-Performance Computing platform (*Loken et al., 2010*).

## Proofs

### Theorem for loss function coordination

The targets that we selected for the hidden layer (see *Equation (8)*) were based on the targets used in *Lee et al., 2015*. The authors of that paper provided a proof showing that their hidden layer targets guaranteed that learning in one layer helped reduce the error in the next layer. However, there were a number of differences between our network and theirs, such as the use of spiking neurons, voltages, different compartments, etc. Here, we modify the original *Lee et al., 2015* proof slightly to prove Theorem 1.

One important thing to note is that the theorem given here utilizes a target for the hidden layer that is slightly different than the one defined in *Equation (8)*. However, the target defined in *Equation (8)* is a numerical approximation of the target given in Theorem 1. After the proof of we describe exactly how these approximations relate to the targets given here.

### Theorem 1

Consider a neural network with one hidden layer and an output layer. Let $\tilde{\boldsymbol{\phi}}^{0*} = \overline{\boldsymbol{\phi}^0}^f + \sigma(Y\overline{\boldsymbol{\phi}^1}^t) - \sigma(Y\phi_{\max}\sigma(E[\overline{V^1}^f]))$ be the target firing rates for neurons in the hidden layer, where $\sigma(\cdot)$ is a differentiable function. Assume that $\overline{V^1}^f \approx k_d \overline{V^{1b}}^f$. Let $\boldsymbol{\phi}^{1*} = \overline{\boldsymbol{\phi}^1}^t$ be the target firing rates for the output layer. Also, for notational simplicity, let $\beta(\boldsymbol{x}) \equiv \phi_{\max}\sigma(k_d W^1 \boldsymbol{x})$ and $\gamma(\boldsymbol{x}) \equiv \sigma(Y\boldsymbol{x})$. Theorem 1 states that if $\boldsymbol{\phi}^{1*} - \phi_{\max}\sigma(E[\overline{V^1}^f])$ is sufficiently small, and the Jacobian matrices $J_\beta$ and $J_\gamma$ satisfy the condition that the largest eigenvalue of $(I - J_\beta J_\gamma)^T (I - J_\beta J_\gamma)$ is less than 1, then

$$||\boldsymbol{\phi}^{1*} - \phi_{\max}\sigma(k_d W^1 \tilde{\boldsymbol{\phi}}^{0*})||_2^2 < ||\boldsymbol{\phi}^{1*} - \phi_{\max}\sigma(E[\overline{V^1}^f])||_2^2$$

We note again that the proof for this theorem is essentially a modification of the proof provided in *Lee et al., 2015* that incorporates our Lemma 1 to take into account the expected value of $\overline{s^0}^f$, given that spikes in the network are generated with non-stationary Poisson processes.

Proof.

$$\phi^{1*} - \phi_{\max}\sigma(k_d W^1 \tilde{\boldsymbol{\phi}}^{0*}) \equiv \phi^{1*} - \beta(\tilde{\boldsymbol{\phi}}^{0*})$$
$$= \phi^{1*} - \beta(\overline{\boldsymbol{\phi}^0}^f + \gamma(\overline{\boldsymbol{\phi}^1}^t) - \gamma(\phi_{\max}\sigma(E[\overline{V^1}^f])))$$

Lemma 1 shows that $\phi_{\max}\sigma(E[\overline{V^1}^f]) = \phi_{\max}\sigma(E[k_d W^1 \overline{s^0}^f]) \approx \phi_{\max}\sigma(k_d W^1 \overline{\phi^0}^f)$ given a sufficiently large averaging time window. Assume that $\phi_{\max}\sigma(E[\overline{V^1}^f]) = \phi_{\max}\sigma(k_d W^1 \overline{\boldsymbol{\phi}^0}^f) \equiv \beta(\overline{\boldsymbol{\phi}^0}^f)$. Then,

$$\phi^{1*} - \beta(\tilde{\boldsymbol{\phi}}^{0*}) = \phi^{1*} - \beta(\overline{\boldsymbol{\phi}^0}^f + \gamma(\overline{\boldsymbol{\phi}^1}^t) - \gamma(\beta(\overline{\boldsymbol{\phi}^0}^f)))$$

Let $\boldsymbol{e} = \overline{\boldsymbol{\phi}^1}^t - \beta(\overline{\boldsymbol{\phi}^0}^f)$. Applying Taylor's theorem,

$$\phi^{1*} - \beta(\tilde{\phi}^{0*}) = \phi^{1*} - \beta(\overline{\phi^{0}}^{f} + J_{\gamma}e + o(||e||_{2}))$$

where $o(||e||_{2})$ is the remainder term that satisfies $\lim_{e \to 0} o(||e||_{2})/||e||_{2} = 0$. Applying Taylor's theorem again,

$$\begin{aligned}
\phi^{1*} - \beta(\tilde{\phi}^{0*}) &= \phi^{1*} - \beta(\overline{\phi^{0}}^{f}) - J_{\beta}(J_{\gamma}e + o(||e||_{2})) \\
&\quad - o(||(J_{\gamma}e + o(||e||_{2}))||_{2}) \\
&= \phi^{1*} - \beta(\overline{\phi^{0}}^{f}) + J_{\beta}J_{\gamma}e - o(||e||_{2}) \\
&= (I - J_{\beta}J_{\gamma})e - o(||e||_{2})
\end{aligned}$$

Then,

$$\begin{aligned}
||\phi^{1*} - \beta(\tilde{\phi}^{0*})||_{2}^{2} &= ((I - J_{\beta}J_{\gamma})e - o(||e||_{2}))^{T}((I - J_{\beta}J_{\gamma})e - o(||e||_{2})) \\
&= e^{T}(I - J_{\beta}J_{\gamma})^{T}(I - J_{\beta}J_{\gamma})e - o(||e||_{2})^{T}(I - J_{\beta}J_{\gamma})e \\
&\quad - e^{T}(I - J_{\beta}J_{\gamma})^{T}o(||e||_{2}) + o(||e||_{2})^{T}o(||e||_{2}) \\
&= e^{T}(I - J_{\beta}J_{\gamma})^{T}(I - J_{\beta}J_{\gamma})e + o(||e||_{2}^{2}) \\
&\leq \mu||e||_{2}^{2} + |o(||e||_{2}^{2})|
\end{aligned}$$

where $\mu$ is the largest eigenvalue of $(I - J_{\beta}J_{\gamma})^{T}(I - J_{\beta}J_{\gamma})$. If $e$ is sufficiently small so that $|o(||e||_{2}^{2}))| < (1 - \mu)||e||_{2}^{2}$, then

$$||\phi^{1*} - \phi_{\max}\sigma(k_{d}W^{1}\tilde{\phi}^{0*})||_{2}^{2} \leq ||e||_{2}^{2} = ||\phi^{1*} - \phi_{\max}\sigma(E[\overline{V^{1}}^{f}])||_{2}^{2}$$

Note that the last step requires that $\mu$, the largest eigenvalue of $(I - J_{\beta}J_{\gamma})^{T}(I - J_{\beta}J_{\gamma})$, is below 1. Clearly, we do not actually have any guarantee of meeting this condition. However, our results show that even though the feedback weights are random and fixed, the feedforward weights actually learn to meet this condition during the first epoch of training (***Figure 5—figure supplement 1***).

## Hidden layer targets

Theorem 1 shows that if we use a target $\tilde{\phi}^{0*} = \overline{\phi^{0}}^{f} + \sigma(Y\overline{\phi^{1}}^{t}) - \sigma(Y\phi_{\max}\sigma(k_{d}W^{1}\overline{\phi^{0}}^{f}))$ for the hidden layer, there is a guarantee that the hidden layer approaching this target will also push the upper layer closer to its target $\phi^{1*}$, if certain other conditions are met. Our specific choice of $\phi^{0*}$ defined in the results (***Equation (8)***) approximates this target rate vector using variables that are accessible to the hidden layer units.

If neuronal units calculate averages after the network has reached a steady state and the firing rates of neurons are in the linear region of the sigmoid function, $\phi_{\max}\sigma(\overline{V^{1}}^{f}) \approx \overline{\phi^{1}}^{f}$. Using Lemma 1, $E[\overline{V^{1}}^{f}] \approx k_{d}W^{1}\overline{\phi^{0}}^{f}$ and $E[\overline{V^{0a}}^{f}] \approx Y\overline{\phi^{1}}^{f}$. If we assume that $\overline{V^{1}}^{f} \approx E[\overline{V^{1}}^{f}]$ and $\overline{V^{0a}}^{f} \approx E[\overline{V^{0a}}^{f}]$, which is true on average, then:

$$\alpha^{f} = \sigma(\overline{V^{0a}}^{f}) \approx \sigma(Y\overline{\phi^{1}}^{f}) \approx \sigma(Y\phi_{\max}\sigma(\overline{V^{1}}^{f})) \approx \sigma(Y\phi_{\max}\sigma(k_{d}W^{1}\overline{\phi^{0}}^{f})) \tag{42}$$

and:

$$\alpha^{t} = \sigma(\overline{V^{0a}}^{t}) \approx \sigma(Y\overline{\phi^{1}}^{t}) \tag{43}$$

Therefore, $\phi^{0*} \approx \tilde{\phi}^{0*}$.

Thus, our hidden layer targets ensure that our model employs a learning rule similar to difference target propagation that approximates the necessary conditions to guarantee error convergence.

### Lemma for firing rates

Theorem 1 had to rely on the equivalence between the average spike rates of the neurons and their filtered spike trains. Here, we prove a lemma showing that this equivalence does indeed hold as long as the integration time is long enough relative to the synaptic time constants $t_s$ and $t_L$.

### Lemma 1

Let $X$ be a set of presynaptic spike times during the time interval $\Delta t = t_1 - t_0$, distributed according to an inhomogeneous Poisson process. Let $N = |X|$ denote the number of presynaptic spikes during this time window, and let $x_k \in X$ denote the $k^{\text{th}}$ presynaptic spike time, where $0 < k \leq N$. Finally, let $\phi(t)$ denote the time-varying presynaptic firing rate (i.e. the time-varying mean of the Poisson process), and $s(t)$ be the filtered presynaptic spike train at time $t$ given by *Equation (11)*. Then, during the time window $\Delta t$, as long as $\Delta t \gg 2\tau_L^2 \tau_s^2 \overline{\phi^2} / (\tau_L - \tau_s)^2 (\tau_L + \tau_s)$,

$$E[\overline{s(t)}] \approx \overline{\phi}$$

Proof. The average of $s(t)$ over the time window $\Delta t$ is

$$\overline{s} = \frac{1}{\Delta t} \int_{t_0}^{t_1} s(t) dt$$

$$= \frac{1}{\Delta t} \sum_k \int_{t_0}^{t_1} \frac{e^{-(t-x_k)/\tau_L} - e^{-(t-x_k)/\tau_s}}{\tau_L - \tau_s} \Theta(t - x_k) dt$$

Since $\Theta(t - x_k) = 0$ for all $t < x_k$,

$$\overline{s} = \frac{1}{\Delta t} \sum_k \int_{x_k}^{t_1} \frac{e^{-(t-x_k)/\tau_L} - e^{-(t-x_k)/\tau_s}}{\tau_L - \tau_s} dt$$

$$= \frac{1}{\Delta t} \left( N - \sum_k \frac{\tau_L e^{-(t_1-x_k)/\tau_L} - \tau_s e^{-(t_1-x_k)/\tau_s}}{\tau_L - \tau_s} \right)$$

The expected value of $\overline{s}$ with respect to $X$ is given by

$$E_X[\overline{s}] = E_X \left[ \frac{1}{\Delta t} \left( N - \sum_k \frac{\tau_L e^{-(t_1-x_k)/\tau_L} - \tau_s e^{-(t_1-x_k)/\tau_s}}{\tau_L - \tau_s} \right) \right]$$

$$= \frac{E_X[N]}{\Delta t} - \frac{1}{\Delta t} E_X \left[ \sum_{k=1}^{N} \left( \frac{\tau_L e^{-(t_1-x_k)/\tau_L} - \tau_s e^{-(t_1-x_k)/\tau_s}}{\tau_L - \tau_s} \right) \right]$$

Since the presynaptic spikes are an inhomogeneous Poisson process with a rate $\phi$, $E_X[N] = \int_{t_0}^{t_1} \phi dt$. Thus,

$$E_X[\overline{s}] = \frac{1}{\Delta t} \int_{t_0}^{t_1} \phi dt - \frac{1}{\Delta t} E_X \left[ \sum_{k=1}^{N} g(x_k) \right]$$

$$= \overline{\phi} - \frac{1}{\Delta t} E_X \left[ \sum_{k=1}^{N} g(x_k) \right]$$

where we let $g(x_k) \equiv (\tau_L e^{-(t_1-x_k)/\tau_L} - \tau_s e^{-(t_1-x_k)/\tau_s})/(\tau_L - \tau_s)$. Then, the law of total expectation gives

$$E_X \left[ \sum_{k=1}^{N} g(x_k) \right] = E_N \left[ E_X \left[ \sum_{k=1}^{N} g(x_k) \Big| N \right] \right]$$

$$= \sum_{n=0}^{\infty} \left( E_X \left[ \sum_{k=1}^{N} g(x_k) \Big| N = n \right] \cdot P(N = n) \right)$$

Letting $f_{x_k}(t)$ denote $P(x_k = t)$, we have that

$$E_X\left[\sum_{k=1}^{N}g(x_k)\bigg|N=n\right]=\sum_{k=1}^{n}E_X[g(x_k)]$$

$$=\sum_{k=1}^{n}\int_{t_0}^{t_1}g(t)f_{x_k}(t)dt$$

Since Poisson spike times are independent, for an inhomogeneous Poisson process:

$$f_{x_k}(t)=\frac{\phi(t)}{\int_{t_0}^{t_1}\phi(u)du}$$

$$=\frac{\phi(t)}{\overline{\phi}\Delta t}$$

for all $t\in[t_0,t_1]$. Since Poisson spike times are independent, this is true for all $k$. Thus,

$$E_X\left[\sum_{k=1}^{N}g(x_k)\bigg|N=n\right]=\frac{1}{\overline{\phi}\Delta t}\sum_{k=1}^{n}\int_{t_0}^{t_1}g(t)\phi(t)dt$$

$$=\frac{n}{\overline{\phi}\Delta t}\int_{t_0}^{t_1}g(t)\phi(t)dt$$

Then,

$$E_X\left[\sum_{k=1}^{N}g(x_k)\right]=\sum_{n=0}^{\infty}\left(\frac{n}{\overline{\phi}\Delta t}\left(\int_{t_0}^{t_1}g(t)\phi(t)dt\right)\cdot P(N=n)\right)$$

$$=\frac{1}{\overline{\phi}\Delta t}\left(\int_{t_0}^{t_1}g(t)\phi(t)dt\right)\left(\sum_{n=0}^{\infty}n\cdot P(N=n)\right)$$

Now, for an inhomogeneous Poisson process with time-varying rate $\phi(t)$,

$$P(N=n)=\frac{\left[\int_{t_0}^{t_1}\phi(t)dt\right]^n e^{-\int_{t_0}^{t_1}\phi(t)dt}}{n!}$$

$$=\frac{[\overline{\phi}\Delta t]^n e^{-(\overline{\phi}\Delta t)}}{n!}$$

Thus,

$$E_X\left[\sum_{k=1}^{N}g(x_k)\right]=\frac{e^{-(\overline{\phi}\Delta t)}}{\overline{\phi}\Delta t}\left(\int_{t_0}^{t_1}g(t)\phi(t)dt\right)\left(\sum_{n=0}^{\infty}n\frac{[\overline{\phi}\Delta t]^n}{n!}\right)$$

$$=\frac{e^{-(\overline{\phi}\Delta t)}}{\overline{\phi}\Delta t}\left(\int_{t_0}^{t_1}g(t)\phi(t)dt\right)(\overline{\phi}\Delta t)e^{\overline{\phi}\Delta t}$$

$$=\int_{t_0}^{t_1}g(t)\phi(t)dt$$

Then,

$$E_X[\overline{s}]=\overline{\phi}-\frac{1}{\Delta t}\left(\int_{t_0}^{t_1}g(t)\phi(t)dt\right)$$

The second term of this equation is always greater than or equal to 0, since $g(t)\geq 0$ and $\phi(t)\geq 0$ for all $t$. Thus, $E_X[\overline{s}]\leq\overline{\phi}$. As well, the Cauchy-Schwarz inequality states that

$$\int_{t_0}^{t_1} g(t)\phi(t)dt \leq \sqrt{\int_{t_0}^{t_1} g(t)^2 dt}\sqrt{\int_{t_0}^{t_1} \phi(t)^2 dt}$$

$$= \sqrt{\int_{t_0}^{t_1} g(t)^2 dt}\sqrt{\overline{\phi^2}\Delta t}$$

where

$$\int_{t_0}^{t_1} g(t)^2 dt = \int_{t_0}^{t_1} \left(\frac{\tau_L e^{-(t_1-t)/\tau_L} - \tau_s e^{-(t_1-t)/\tau_s}}{\tau_L - \tau_s}\right)^2 dt$$

$$\leq \frac{1}{2(\tau_L - \tau_s)^2}\left(4\frac{\tau_L^2\tau_s^2}{\tau_L + \tau_s}\right)$$

$$= \frac{2\tau_L^2\tau_s^2}{(\tau_L - \tau_s)^2(\tau_L + \tau_s)}$$

Thus,

$$\int_{t_0}^{t_1} g(t)\phi(t)dt \leq \sqrt{\frac{2\tau_L^2\tau_s^2}{(\tau_L - \tau_s)^2(\tau_L + \tau_s)}}\sqrt{\overline{\phi^2}\Delta t}$$

$$= \sqrt{\Delta t}\sqrt{\frac{2\tau_L^2\tau_s^2\overline{\phi^2}}{(\tau_L - \tau_s)^2(\tau_L + \tau_s)}}$$

Therefore,

$$E_X[\bar{s}] \geq \overline{\phi} - \frac{1}{\Delta t}\sqrt{\Delta t}\sqrt{\frac{2\tau_L^2\tau_s^2\overline{\phi^2}}{(\tau_L - \tau_s)^2(\tau_L + \tau_s)}}$$

$$= \overline{\phi} - \sqrt{\frac{2\tau_L^2\tau_s^2\overline{\phi^2}}{\Delta t(\tau_L - \tau_s)^2(\tau_L + \tau_s)}}$$

Then,

$$\overline{\phi} - \sqrt{\frac{2\tau_L^2\tau_s^2\overline{\phi^2}}{\Delta t(\tau_L - \tau_s)^2(\tau_L + \tau_s)}} \leq E_X[\bar{s}] \leq \overline{\phi}$$

Thus, as long as $\Delta t \gg 2\tau_L^2\tau_s^2\overline{\phi^2}/(\tau_L - \tau_s)^2(\tau_L + \tau_s)$, $E_X[\bar{s}] \approx \overline{\phi}$.

What this lemma says, effectively, is that the expected value of $s$ is going to be roughly the average presynaptic rate of fire as long as the time over which the average is taken is sufficiently long in comparison to the postsynaptic time constants and the average rate-of-fire is sufficiently small. In our simulations, $\Delta t$ is always greater than or equal to 50 ms, the average rate-of-fire is approximately 20 Hz, and our time constants $\tau_L$ and $\tau_s$ are 10 ms and 3 ms, respectively. Hence, in general:

$$2\tau_L^2\tau_s^2\overline{\phi^2}/(\tau_L - \tau_s)^2(\tau_L + \tau_s) = 2(10)^2(3)^2(0.02)^2/(10-3)^2(10+3)$$

$$\approx 0.001$$

$$\ll 50$$

Thus, in the proof of Theorem 1, we assume $E_X[\bar{s}] = \overline{\phi}$.

## Acknowledgements

We would like to thank Douglas Tweed, João Sacramento, and Yoshua Bengio for helpful discussions on this work. This research was supported by three grants to BAR: a Discovery Grant from the Natural Sciences and Engineering Research Council of Canada (RGPIN-2014–04947), a 2016 Google Faculty Research Award, and a Fellowship with the Canadian Institute for Advanced Research. The

authors declare no competing financial interests. Some simulations were performed on the gpc supercomputer at the SciNet HPC Consortium. SciNet is funded by: the Canada Foundation for Innovation under the auspices of Compute Canada; the Government of Ontario; Ontario Research Fund - Research Excellence; and the University of Toronto.

## Additional information

### Funding

| Funder | Grant reference number | Author |
|---|---|---|
| Natural Sciences and Engineering Research Council of Canada | RGPIN-2014-04947 | Blake A Richards |
| Google | Faculty Research Award | Blake A Richards |
| Canadian Institute for Advanced Research | Learning in Machines and Brains Program | Blake A Richards |

The funders had no role in study design, data collection and interpretation, or the decision to submit the work for publication.

### Author contributions

Jordan Guerguiev, Conceptualization, Data curation, Software, Formal analysis, Visualization, Methodology, Writing—original draft, Project administration, Writing—review and editing; Timothy P Lillicrap, Conceptualization, Methodology, Writing—original draft, Writing—review and editing; Blake A Richards, Conceptualization, Resources, Supervision, Funding acquisition, Methodology, Writing—original draft, Project administration, Writing—review and editing

### Author ORCIDs

Jordan Guerguiev (ID) http://orcid.org/0000-0002-6751-8782
Blake A Richards (ID) http://orcid.org/0000-0001-9662-2151

### Decision letter and Author response

Decision letter https://doi.org/10.7554/eLife.22901.026
Author response https://doi.org/10.7554/eLife.22901.027

## Additional files

### Supplementary files

• Transparent reporting form
DOI: https://doi.org/10.7554/eLife.22901.022

### Major datasets

The following previously published dataset was used:

| Author(s) | Year | Dataset title | Dataset URL | Database, license, and accessibility information |
|---|---|---|---|---|
| LeCun Y, Bottou L, Bengio Y, Haffner P | 1998 | MNIST | http://yann.lecun.com/exdb/mnist/ | Publicly available at yann.lecun.com |

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
