## [Decision Letter]

[Editors’ note: this article was originally rejected after discussions between the reviewers, but the authors were invited to resubmit after an appeal against the decision.]

Thank you for submitting your work entitled "Deep learning with segregated dendrites" for consideration by *eLife*. Your article has been evaluated by a Senior Editor and three reviewers, one of whom is a member of our Board of Reviewing Editors. The reviewers have opted to remain anonymous.

Our decision has been reached after consultation among the reviewers. Based on these discussions, which are summarized below together with the individual reviews, we regret to inform you that your work will not be considered further for publication in *eLife*.

Reviewing Editor's summary:

This was a tough one: reviewers 2 and 3 were very positive, and even reviewer 1, who was negative about the clarity of the paper, was very positive about its content and importance. The problem was the writing: the reviewers felt that in its present form, the paper would be understandable only by deep learning experts. I'm sure it would be possible to fix this, but the reviewers also felt that this would be a major undertaking, and might even take a couple of rounds. It is *eLife*'s policy to reject papers in that situation.

I'm very sorry; I would love to see work like this in *eLife*. Unfortunately, I'm not sure how useful the reviews will be – the most negative reviewer was #1, but there were only a small number of concrete suggestions, mainly because s/he was very lost. Perhaps it would be helpful to find a theoretical neuroscientist who is not an expert in deep networks – presumably your target audience – and see where s/he has trouble understanding the paper.

*Reviewer #1:*

This paper touches on a very important topic: biologically plausible deep learning. However, this particular version is not suitable for *eLife*. In fact, it's not clear it's suitable at all: after several hours staring at the paper, I remained thoroughly confused. Please don't get me wrong; I'm guessing the paper is correct; I think the problem is mainly the exposition relative to my level of knowledge.

A few examples:

1) It was never clear from the notation (and often the text) whether they were referring to scalars, vectors or matrices – something that does not help when one is trying to make sense of the math.

2) Above Equation (1) the authors talk about target firing rates. But, except for the output units, it's not clear at all what those are.

3) In Equation (1), I don't know what the target burst, α^t, is. I thought the apical dendrite (presumably what α is referring to) is cut off in the target phase.

4) Why should Equation (1) be the target rate?

5) L^0 is actually |α^t-α^f|^2. Why not say so?

At this point I turned to Materials and methods, in the hopes that the equations would clarify things. They didn't.

6) Equations (5)-(8) are standard, but are written in a very complicated a form. There may be a reason for that, but it's confusing for your run of the mill computational neuroscientist.

7) As far as I could tell, neither Equations (7) nor (8) include the feedback from the apical dendrite. And I couldn't figure out from anywhere in Materials and methods how that was implemented.

8) Equation (17) seems inconsistent with Equations (19) and (20).

And at that point I gave up.…

*Reviewer #2:*

This paper takes on the valiant task of making artificial deep neural networks more biologically relevant by implementing multi-compartmental neurons. In particular, the segregation of feed-forward from feedback information processing streams within the single cell is a welcome addition for biologists to see in computational models. The authors use details about the anatomical and physiological properties of cortical pyramidal neurons to implement backprop training. They establish that these biologically-inspired features can be accommodated without significant loss of performance. We believe this paper would be a welcome early step in the direction of bringing deep artificial network and neurophysiology thinking together, but requires conceptual explanation in a few key areas, especially for biologists not familiar with the details of deep networks.

What is the conceptual reason that feedforward and feedback streams need to be separated? Is it because the error signal is computed as the difference between the forward phase and the "correct" answer imposed by the teaching signal on the output neurons in the target phase? Conceptually, it seems that the separation of the signals allows for an error to be computed, and therefore for the appropriate change in weights to be arrived at. This is in contrast to how some often think about the relationship between feedforward and feedback in the brain where the main function of the feedforward/feedback integration is to actively and directly create downstream activity (as opposed to here where it is to change the weights of synapses).

What is the purpose of the random sampling of bursts? Why not just a fixed time? Would asynchronous bursting still be effective? Is the synchronous nature of the bursting in order to coordinate with the feedback from the teaching signal?

Would all of this be mathematically equivalent to a separate set of neurons that deal primarily with teaching signals in feedback pathways, and whose interaction with the "normal" feedforward network be regulated through some disinhibitory mechanism? To say this another way, is there anything special about the single cells and the nonlinearity α used, or could a similar setup be created by separating the different compartments into single neurons and connecting them with normal synapses?

What is the explanation for why weak apical attenuation disrupts learning? Is it because it forces an underestimation of the error by having the difference in activity between forward and target phases become eroded?

Local here means local in space. However, in order to compute weight updates, differences in activity still need to be taken over time. More specifically, the activity in the bursts between forward and target phases (equation 2). What is the biologically plausible mechanism for such non-temporally aligned computation?

Is there any explanation for why sparse feedback weights improve the network?

In general it would useful to have conceptual explanations for many of the issues discussed above.

*Reviewer #3:*

I think this is a very valuable manuscript that makes a link between deep learning and a possible biological implementation. As this link is of high scientific relevance topic and of broad interest, I consider the manuscript suited for a good journal as *eLife*, even if there is still a large gap between the performance of deep learning for artificial neuronal network and the suggest biological implementation (that only considers 2 layers with relatively humble performance). But the authors well recognize this and the manuscript represents a first step towards future research in this important field.

There is one main issue that should be addressed more thoroughly.

1) The proof of Theorem 1 assumes that the matrix product (J_β) (J_γ) is close to the identity mapping in the readout space. In the cited work by Lee et al. (Difference Propagation, 2015) this is the case because the forward and backward weights are adapted such that they get aligned. In the present case the alignment only becomes indirectly apparent by simulations showing that the error vector in the hidden layer eventually falls within 90 degrees of the true backpropagation error.

As I understand, the top-down weight matrix Y is fixed (e.g. randomly chosen). From a theoretical perspective, one may choose Y to be the pseudo-inverse of the forward weight matrix W^1. In fact, in that case a much simpler proof for Theorem 1 exists (a few lines only). But if Y is random, then the whole idea boils down to the random feedback idea (Lillicrap et al., Nature communication 2016) and this link should be emphasized more. While in the Supplementary Information of that paper a proof is outlined for linear transfer functions, it remains unclear how for nonlinear transfer functions this alignment is achieved obtained.

If J is chosen to be the transposed of W^1 as it is the case in backprop (and in part of the simulations), then nothing has to be proven. But if Y is random, then the big issue is to prove that the mapping γ(y) is approximatively an inversion of the mapping β(x). If this were proven, Theorem 1 in the manuscript could be cited as Theorem 2 in Lee et al. (Diff prop, 2015). But in the current form, Theorem 1 replicates the idea of Lee et al. (as it is also stated by the authors) without proving the basic assumption shown to be true in the case of Lee et al.. Of course, for the reader's convenience the proof of the Diff-Prop Theorem can still be reproduced.

In my view the core idea for the theory in the paper is (1) with random top-down connections the forward weights align as shown by Lillicrap et al. (2) Given the alignment, the idea of difference propagation with the proof given in Lee et al. can be applied. Once this theoretical fundament is introduced in this form (and simply referred to these papers), the idea of using segregated dendrites to implement the random feedback idea can be stressed.

A bit less fundamental, but still more than minor:

2) In view of the rather deep mathematical issues related to the feedback alignment, I would suggest to defer Lemma 1 to some Supplementary Information. The approximation of PSP signaling by instantaneous Poisson rates when the rate is small as compared to the PSP duration is standard in theoretical neuroscience. But the 3-page proof is still nicely done and may be helpful for a non-specialist who wishes to go into the details.

3) At the end of the subsection “A network architecture with segregated dendritic compartments” (Results) some critical issues are raised about the biological plausibility. In this context it should also be stressed that the alternation between two phases, each of which again subdivided into two further phases (Figure 1), is not so easy to match to the biology. The phases need a memory that is tagged with the phase information and plasticity that is only turned on in a specific phase, checking out the memory from a previous phase.

Beside mentioning this in the Results, it should also be taken up in a further paragraph in the Discussion. One should mention that synaptic eligibility traces could help out here and that this helps to bridge information across the phases. Moreover, the phases could be implemented by exploiting global (I guess γ) oscillations that are shown to be present in various cognitive states. Discussing the link of learning and γ oscillations may be of general interest in this context.

[Editors’ note: what now follows is the decision letter after the authors submitted for further consideration.]

Thank you for resubmitting your work entitled "Towards deep learning with segregated dendrites" for further consideration at *eLife*. Your article has been favorably evaluated by Andrew King (Senior Editor) and three reviewers, one of whom is a member of our Board of Reviewing Editors.

This paper is much improved. However, it still has a way to go before it's ready for a neuroscience audience. Given that this has been reviewed several times now and remains in an unacceptable form, we are prepared to offer only one more opportunity to provide an acceptable version of the manuscript.

The easy thing to fix is notation and writing: we believe that, even in its improved form, it would be very hard for a neuroscientist, even a computational one who is used to thinking about circuits, to read, and the main ideas would be difficult to extract. More on that below.

The potentially harder thing to fix is biological plausibility. If we understand things correctly, the neuron must estimate the average PSPs during the feedforward sweep of activity, when only the input is active, estimate them again during the training phase, when the correct output is active as well, and then subtract the two. These signals are separated in time, which means the synapses have to store one signal, wait a few tens of ms, store another, and take the difference. In addition, the difference is computed in the apical dendrite, but it must be transferred to the proximal dendrites. And finally, a global signal is required to tell a synapse in which phase it is so that the estimate can be endowed with the correct sign. All seem nontrivial for neurons and synapses to compute.

Lack of biological plausibility should not rule out a theory – synapses and neurons are, after all, complicated. However, two things are needed. First, you need to provide a mechanism for implementing the learning rule that's not inconsistent with what's known about neurons and synapses. Second, you need to provide suggest experiments to test these predictions. Of these, the first is probably harder.

Now for the exposition. It may seem like we're micromanaging (and we are), but if this paper is to have an impact on the neuroscience community – a prerequisite for publication in *eLife* – it has to be cast into familiar notation.

1) The model was much simpler, and more standard, than first impressions would imply. It can be written in the very familiar form

dV^m_i/dt = -g_L V^m_i

+ sum_n g_n (b^n_i + sum_j W^mn_ij s^n_j(t) – V^m_i)

+ g^m_iE(t) (E_E – V_i^m) + g^m_iI(t) (E_I – V_i^m)

where the s^n_j(t) are filtered spike trains,

s^n_j(t) = sum_k kappa(t-t^n_jk),

t^n_jk is the k^th spike on neuron j of type n, and spikes were generated via a Poisson process based on the voltage. (Please note: errors are possible, but the equations look something like what we wrote.)

In this form it is immediately clear to a neuroscientist what kind of network this is. If nothing else, that will save a huge amount of time for the reader – it took hours of going back and forth over the equations in the paper before it became clear that the model was very standard, something that most readers would not have the patience for.

In addition, as written above, it makes it clear exactly how the dendrites are implemented: by varying the g_n. In real dendrites they vary with voltage on the dendrite; in this model they simply vary with time.

And finally, the notation with the A's and B's used in the paper is not helpful to neuroscientists, who are very used to seeing V, or maybe U, for voltage.

2) Along the same lines, a better figure showing the circuit needs to be included. The circuit with multiple hidden layers needs a similar drawing, as we were not able to figure out exactly what it looked like. (We're guessing there was sufficient information in the paper, but the amount of work it would take to extract it seemed high.)

3) The cost function, L^1, seemed somewhat arbitrary. According to Equation 7,

L^1 \propto \sum_i (<σ(U_i)>^t – σ(<U_i>^f))^2

where the angle brackets represent a time average and the superscripts t and f refer to the target and feedforward phases, respectively (basically, the overline was replaced with angle brackets, mainly because we're using plain text). Why was the average taken outside the sigmoid in the target phase and inside the sigmoid in the feedforward phase?

4) A similar question applies to L^0, which is written

L^0 = sum_i (λ_max(<σ(C_i)>^f – σ(<C_i>^f) + α_i^t – α_i^f)^2

As far as we can tell, the first two terms are included to make the update rules work out, and they are eventually set equal to each other. But is there any reason to think that L^0 should be minimized? It seemed unmotivated.

5) In Equations 19 and 22, why are there no terms involving the derivatives of the sigmoid in the target phase?

---

## [Author Response]

[Editors’ note: the author responses to the first round of peer review follow.]

Reviewing Editor's summary:This was a tough one: reviewers 2 and 3 were very positive, and even reviewer 1, who was negative about the clarity of the paper, was very positive about its content and importance. The problem was the writing: the reviewers felt that in its present form, the paper would be understandable only by deep learning experts. I'm sure it would be possible to fix this, but the reviewers also felt that this would be a major undertaking, and might even take a couple of rounds. It is eLife's policy to reject papers in that situation.I'm very sorry; I would love to see work like this in eLife. Unfortunately, I'm not sure how useful the reviews will be – the most negative reviewer was #1, but there were only a small number of concrete suggestions, mainly because s/he was very lost. Perhaps it would be helpful to find a theoretical neuroscientist who is not an expert in deep networks – presumably your target audience – and see where s/he has trouble understanding the paper.Reviewer #1:This paper touches on a very important topic: biologically plausible deep learning. However, this particular version is not suitable for eLife. In fact, it's not clear it's suitable at all: after several hours staring at the paper, I remained thoroughly confused. Please don't get me wrong; I'm guessing the paper is correct; I think the problem is mainly the exposition relative to my level of knowledge.

We would like to thank reviewer #1 for their comments. We are very pleased that the reviewer recognizes that this is a “…very important topic”. Importantly, we also agree with reviewer #1 that our manuscript, as written, was not pitched at the appropriate level. This was a critical realization for us, and it has helped us to make the paper far more suitable for the general readership of *eLife*. With some advice from non-specialist readers, we have done a major re-write of the manuscript, especially the early parts where we introduce the central issues and describe our model. As the reviewer will see, we have completely re-written the Introduction and the first half of the Results. As well, we have included two new introductory figures (see Figure 1 and Figure 2) that lay out the issues and describe our approach to solving them in a manner that we believe will be much easier for a general audience to understand. In particular, we now do the following:

1) We define the “credit assignment problem”, and we explain why it is important for neuroscientists to consider. This was, arguably, a major missing piece of explanation in our original submission. Readers who are unfamiliar with deep learning may not have considered the fact that effective synaptic plasticity rules in a multi-layer/multi-circuit network will require some way for neurons to know something about their contribution to the final output. The Introduction and Figure 1 now describe this issue in a manner that is generally accessible. As well, in the Introduction and Figure 1, we also describe how the backpropagation of error algorithm solves the credit assignment problem with “weight transport”.

2) We provide a concrete explanation of how we are proposing to solve the credit assignment problem. In particular, in the Introduction and Figure 2 we now clarify that: (1) one key to assigning credit in current deep learning models is keeping separate feedforward and feedback calculations, and (2) the main goal of this paper is to accomplish this separation of feedforward and feedback signals in a biologically feasible manner that does not involve a separate feedback pathway, as is implicitly assumed in previous models (such as Lillicrap et al., 2016 and Lee et al., 2015).

3) We provide a more comprehensible description of our model in the first section of the Results. In particular, we now clarify which variables refer to vectors, and which variables refer to scalar values (in fact, we have now adopted a notation where vectors and matrices are always in boldface). Furthermore, when we describe the dynamics of the neurons we do so using the equations for single neurons, and we use a more commonplace notation for differential equations. Finally, we are also careful to fully define all of the values that appear in our target and loss function equations in the Results, as these are key to understanding how the algorithm works.

With the new figures, new notation, and re-written manuscript, we believe that the paper is now much easier for all readers to understand. We hope that reviewer #1 agrees. Below, we address some of reviewer #1's specific comments.

A few examples:1) It was never clear from the notation (and often the text) whether they were referring to scalars, vectors or matrices – something that does not help when one is trying to make sense of the math.

We thank reviewer #1 for drawing our attention to this point. Other readers have also told us that it was very easy for those who are unfamiliar with deep learning algorithms to get lost in our original use of scalars, versus vectors or matrices, especially when it was not explicitly stated. To help this we have done three things. First, we have adopted a notation where vectors and matrices are always in boldface, and scalars are not. Second, we have been careful to always define our vectors to make clear which scalar values they contain. For example, when we define the somatic voltage vector now, we refer to “**C**(*t*) = [*C*_1_(*t*),.…, *C_m_(t*)]” (see e.g. subsection “A network architecture with segregated dendritic compartments”, fourth paragraph). Third, to make the model easier to understand, we have attempted to define the dynamics of the model in terms of the individual scalar variables rather than the vectors whenever possible (see e.g. Equation (1)). We believe that these changes have significantly improved the readability of the paper.

2) Above Equation (1) the authors talk about target firing rates. But, except for the output units, it's not clear at all what those are.

We agree with reviewer #1 that this was unclear previously. We now spend much more time explicitly defining the target firing rates. For example, we now state in the Results:

“…we defined local targets for the output and the hidden layer, i.e. desired firing rates for both the output layer neurons and the hidden layer neurons. Learning is then a process of changing the synaptic connections to achieve these target firing rates across the network.”,.

For both the output and the hidden layer neurons we provide more explanation of the target rates that we define. In the Results, using Equations (4), (5) and (6), we are now careful to define the target firing rates on both a mathematical and conceptual level. For example, for the hidden layer targets, we state:

“For the hidden layer we define the target rates-of-fire… using the average rates-of-fire during the forward phase and the difference between the plateau potentials from the forward and transmit phase… The goal of learning in the hidden layer is to change the synapses **W**^0^ to achieve these targets in response to the given inputs.”

We also provide the reasoning motivating these hidden targets when we discuss the loss functions (see the responses to points 4 and 5 below). Also, we are now careful to define all of the components of our equations before we use them (see Equation (4)). We think that this is a major improvement on the original manuscript, and key to making the paper more enjoyable to read. We hope that reviewer #1 agrees.

3) In Equation (1), I don't know what the target burst, α^t, is. I thought the apical dendrite (presumably what α is referring to) is cut off in the target phase.

This is a perfect example of the lack of clarity in our original manuscript. Again, we thank the reviewer for drawing our attention to this. We have attempted to address this in two ways. First, in order to be more transparent about what these **α** values actually are, we have renamed them “plateau potentials”. The reason we do that is that they are actually just non-linear versions of the apical dendrite voltages, rather than actual bursts of spikes. Second, we now define the forward and target “plateau potentials” (**α**^t^ and **α**^f^) explicitly (see subsection “A network architecture with segregated dendritic compartments”, eighth paragraph and Equation (3)).

4) Why should Equation (1) be the target rate?

We now try to provide a more intuitive explanation for this in the subsection “Credit assignment with segregated dendrites”. Please see the next point for more description.

5) L^0 is actually |α^t-α^f|^2. Why not say so?

Indeed, the reviewer is correct, L^0^ will, on average, reduce to ||**α**^t^-**α**^f^||^2^, and we should have said so. We now state this in the text (subsection “Credit assignment with segregated dendrites”, sixth paragraph). Furthermore, we also point out that this reduction helps to explain why our target, as defined, helps with credit assignment. Specifically, with an error function equal to ||**α**^t^-**α**^f^||^2^, we ensure that when the output layer is sending the same feedback to the hidden layer during both the forward and target phases, then the hidden layer neurons know that they have converged to appropriate representations for accomplishing the categorization task.

Now, as the reviewer asks, the question is, why not simply do this reduction? Why define L^0^ as we have? The reason is twofold. First, the reduction is not actually 100% accurate on any given trial, since the average rates-of-fire of the neurons (λ_i_) are not necessarily exactly the same thing as the rate-of-fire that one would get if one applied the sigmoid function to the average voltage (σ(C_i_(t))). Second, the reduction makes it appear as if L^0^ was not a function of the hidden layer activity. But it is, and this is key to calculating the gradient of the loss function with respect to the hidden layer synapses, W^0^ (see Equation (22)).

At this point I turned to Materials and methods, in the hopes that the equations would clarify things. They didn't.6) Equations (5)-(8) are standard, but are written in a very complicated a form. There may be a reason for that, but it's confusing for your run of the mill computational neuroscientist.

The form of equation we used originally is common in a number of fields, but we have replaced it with a more standard format that is typical in computational neuroscience (see Equation (1)). This is more appropriate given the target audience of this paper, so we thank the reviewer for pointing this out.

7) As far as I could tell, neither Equations (7) nor (8) include the feedback from the apical dendrite. And I couldn't figure out from anywhere in Materials and methods how that was implemented.

In the original manuscript, Equation (7) did not include apical feedback, but Equation (8) did. We have replaced both equations with Equation (1) in the new manuscript, and stated clearly how we determine the level of apical feedback (i.e. using g_A_, see subsection “A network architecture with segregated dendritic compartments”, fourth paragraph).

8) Equation (17) seems inconsistent with Equations (19) and (20).

We are not sure why reviewer #1 felt that these equations were inconsistent. In the new version of the manuscript we have attempted to place these equations in a more appropriate location that makes their relevance more obvious.

In summary, we truly are indebted to reviewer #1 for identifying the lack of clarity in our original manuscript, and we recognize that it was not sufficiently accessible. But, we feel that with our major re-write, new figures, and new notation the paper is now well-suited to the readership of *eLife*. We want to emphasize that, with this paper, our goal is to get physiologists and computational neuroscientists to think differently about the reasons for pyramidal neuron morphology/physiology. That is why we feel it is important for it to be published in a journal with a broad readership, like *eLife*, rather than in a more specialist journal. Thanks to reviewer #1, we feel that our paper can now achieve these goals. We hope reviewer #1 agrees.

And at that point I gave up.…Reviewer #2:This paper takes on the valiant task of making artificial deep neural networks more biologically relevant by implementing multi-compartmental neurons. In particular, the segregation of feed-forward from feedback information processing streams within the single cell is a welcome addition for biologists to see in computational models. The authors use details about the anatomical and physiological properties of cortical pyramidal neurons to implement backprop training. They establish that these biologically-inspired features can be accommodated without significant loss of performance. We believe this paper would be a welcome early step in the direction of bringing deep artificial network and neurophysiology thinking together, but requires conceptual explanation in a few key areas, especially for biologists not familiar with the details of deep networks.

We are very happy that reviewer #2 recognizes that “…making artificial deep neural networks more biologically relevant by implementing multi-compartmental neurons.” is a valiant task, and that they view our paper as “…a welcome early step in the direction of bringing deep artificial network and neurophysiology thinking together…” We agree with the reviewer that there are a number of conceptual issues that required clarification. Below, we address reviewer #2's specific comments.

What is the conceptual reason that feedforward and feedback streams need to be separated? Is it because the error signal is computed as the difference between the forward phase and the "correct" answer imposed by the teaching signal on the output neurons in the target phase? Conceptually, it seems that the separation of the signals allows for an error to be computed, and therefore for the appropriate change in weights to be arrived at. This is in contrast to how some often think about the relationship between feedforward and feedback in the brain where the main function of the feedforward/feedback integration is to actively and directly create downstream activity (as opposed to here where it is to change the weights of synapses).

This is a key issue in our paper, and we are very grateful that reviewer #2 requested more conceptual explanation. First, we are now very clear about why feedforward information must be integrated separately from feedback information for this form of deep learning algorithm to work. We now state:

“…synaptic weight updates in the hidden layers (of previous models) depend on the difference between feedback that is generated in response to a purely feedforward propagation of sensory information, and feedback that is guided by a teaching signal (Lillicrap et al., 2016; Lee et al., 2015; Liao et al., 2015). In order to calculate this difference, sensory information must be transmitted separately from the feedback signals that are used to drive learning.”

This provides the reason for segregating feedback in apical dendrites. As the reviewer points out though, this way of viewing feedback (as a signal to drive learning, rather than a higher-order modulator of low-level activity), is not common in neuroscience. However, the two potential roles for feedback are not necessarily incompatible (as noted in previous models like Spratling and Johnson, 2006 and Körding and König, 2002). Our model focuses on the role of feedback in learning exclusively, but it is likely that future researchers will find ways of combining these functions in deep learning networks. To that end, we have added the following to the Discussion with new references:

“…framing cortico-cortical feedback as a mechanism to modulate incoming sensory activity is a more common way of viewing feedback signals in the neocortex (Larkum, 2013; Gilbert and Li, 2013; Zhang et al. 2014; Fiser et al. 2016). […] Future studies could examine how top-down modulation and a signal for credit assignment can be combined in deep learning models.”

What is the purpose of the random sampling of bursts? Why not just a fixed time? Would asynchronous bursting still be effective? Is the synchronous nature of the bursting in order to coordinate with the feedback from the teaching signal?

This is an excellent question. We ourselves were not sure of the answer immediately. Based on the definitions we give, our intuition was that explicit synchrony was not required, though the temporal relationship between the bursts/plateau potentials and the teaching signal would be important. (Note: we have renamed the “bursts” as “plateau potentials” in this version in order to make their actual form more transparent.) To determine this, we ran some simulations wherein each hidden layer neuron sampled its own inter-plateau interval during each phase, and we examined whether this affected learning. We found that strict synchrony was not, in fact, required and learning proceeded just as well with neurons engaging in plateau potentials at different times (Figure 5—figure supplement 1). However, learning would undoubtedly not work if the teaching signal input was not straddled by the two different plateau potentials. We now note this in the text:

“…(learning was still) obtained when we loosened the synchrony constraints and instead allowed each hidden layer neuron to engage in plateau potentials at different times (Figure 5—figure supplement 1). This demonstrates that strict synchrony in the plateau potentials is not required. But, our target definitions do require two different plateau potentials separated by the teaching signal input, which mandates some temporal control of plateau potentials in the system.” –Would all of this be mathematically equivalent to a separate set of neurons that deal primarily with teaching signals in feedback pathways, and whose interaction with the "normal" feedforward network be regulated through some disinhibitory mechanism? To say this another way, is there anything special about the single cells and the nonlinearity α used, or could a similar setup be created by separating the different compartments into single neurons and connecting them with normal synapses?

Indeed, the reviewer's intuition is 100% correct: we could accomplish the same error signal we use to learn in the hidden layers using a separate feedback pathway, thereby replacing our apical dendritic compartments with other neurons. We now explicitly state this (Introduction, sixth paragraph) and even provide an introductory figure that highlights this other potential solution (Figure 2).

In fact, in order to make the motivations for the paper more obvious, we now spend some of the Introduction discussing why we are inclined to explore an alternative to a separate feedback pathway:

“…closer inspection uncovers a couple of difficulties with (using a separate feedback pathway)… First, the error signals that solve the credit assignment problem are not global error signals (like neuromodulatory signals used in reinforcement learning). […] Therefore, the real brain's specific solution to the credit assignment problem is unlikely to involve a separate feedback pathway for cell-by-cell, signed signals to instruct plasticity.”

What is the explanation for why weak apical attenuation disrupts learning? Is it because it forces an underestimation of the error by having the difference in activity between forward and target phases become eroded?

The reason that weak apical attenuation disrupts learning is precisely that it prevents the feedback regarding the forward phase (**α**^f^) from cleanly communicating the output that the feedforward information generated. We now state this:

“This demonstrates that although total apical attenuation is not necessary, partial segregation of the apical compartment from the soma is necessary. […] Our data show that electrontonically segregated dendrites is one potential way to achieve the required separation between feedforward and feedback information.”

Local here means local in space. However, in order to compute weight updates, differences in activity still need to be taken over time. More specifically, the activity in the bursts between forward and target phases (equation 2). What is the biologically plausible mechanism for such non-temporally aligned computation?

This is a fantastic question. In some ways, our algorithm trades spatial non-locality for temporal nonlocality. However, the temporal non-locality in the network is relatively small (e.g. voltage and/or plateau potential information must be stored for tens of milliseconds), which could potentially be implemented with molecular mechanisms, such as synaptic tags (Redondo and Morris, 2011). We now make this temporal non-locality explicit:

“It should be recognized, though, that although our learning algorithm achieved deep learning with spatially local update rules, we had to assume some temporal non-locality. […] Hence, our model exhibited deep learning using only local information contained within the cells.”Is there any explanation for why sparse feedback weights improve the network?

The reviewer asks another great question here. Again, we were unsure of the answer at first. In exploring the effects of sparse feedback further, we found that the issue may be one of the scale of feedback weights. Specifically, when we ran the tests on sparse feedback weights in the original manuscript we increased the magnitude of the weights 5x (since we were eliminating 80% of the weights). However, following on this question from reviewer #2, we explored sparse feedback weights without the 5x re-scaling. In this case, we found that learning was impaired (Figure 7—figure supplement 1). Thus, we believe that sparse feedback itself is not beneficial, rather the real reason that sparse feedback weights improved learning in the network was that we were amplifying the difference signals. We now discuss this in the results and include a supplementary figure with this data:

“We found that learning actually improved slightly with sparse weights (Figure 7, red line), achieving an average error rate of 3.7% by the 60th epoch, compared to the average 4.1% error rate achieved with fully random weights. […] This suggests that sparse feedback provides a signal that is sufficient for credit assignment, but only if it is of appropriate magnitude.”

In general it would useful to have conceptual explanations for many of the issues discussed above.Reviewer #3:I think this is a very valuable manuscript that makes a link between deep learning and a possible biological implementation. As this link is of high scientific relevance topic and of broad interest, I consider the manuscript suited for a good journal as eLife, even if there is still a large gap between the performance of deep learning for artificial neuronal network and the suggest biological implementation (that only considers 2 layers with relatively humble performance). But the authors well recognize this and the manuscript represents a first step towards future research in this important field.

We are pleased that reviewer #3 recognizes that “…this is a very valuable manuscript that makes a link between deep learning and a possible biological implementation…” and that “… this link is of high scientific relevance topic and of broad interest…” and therefore well-suited to publication in *eLife*. As the reviewer points out, there is still a large gap between deep learning in artificial neural networks and our understanding of the neurobiology of learning, and like the reviewer, we also believe that this manuscript “… represents a first step towards future research in this important field.” We found reviewer #3's criticisms to be very constructive, and we feel that we have addressed each of their concerns. Below, we address reviewer #3's specific comments.

There is one main issue that should be addressed more thoroughly.1) The proof of Theorem 1 assumes that the matrix product (J_β) (J_γ) is close to the identity mapping in the readout space. In the cited work by Lee et al. (Difference Propagation, 2015) this is the case because the forward and backward weights are adapted such that they get aligned. In the present case the alignment only becomes indirectly apparent by simulations showing that the error vector in the hidden layer eventually falls within 90 degrees of the true backpropagation error.As I understand, the top-down weight matrix Y is fixed (e.g. randomly chosen). From a theoretical perspective, one may choose Y to be the pseudo-inverse of the forward weight matrix W^1. In fact, in that case a much simpler proof for Theorem 1 exists (a few lines only). But if Y is random, then the whole idea boils down to the random feedback idea (Lillicrap et al., Nature communication 2016) and this link should be emphasized more. While in the Supplementary Information of that paper a proof is outlined for linear transfer functions, it remains unclear how for nonlinear transfer functions this alignment is achieved obtained.If J is chosen to be the transposed of W^1 as it is the case in backprop (and in part of the simulations), then nothing has to be proven. But if Y is random, then the big issue is to prove that the mapping γ(y) is approximatively an inversion of the mapping β(x). If this were proven, Theorem 1 in the manuscript could be cited as Theorem 2 in Lee et al. (Diff prop, 2015). But in the current form, Theorem 1 replicates the idea of Lee et al. (as it is also stated by the authors) without proving the basic assumption shown to be true in the case of Lee et al.. Of course, for the reader's convenience the proof of the Diff-Prop Theorem can still be reproduced.In my view the core idea for the theory in the paper is (1) with random top-down connections the forward weights align as shown by Lillicrap et al. (2) Given the alignment, the idea of difference propagation with the proof given in Lee et al. can be applied. Once this theoretical fundament is introduced in this form (and simply referred to these papers), the idea of using segregated dendrites to implement the random feedback idea can be stressed.

Reviewer #3 has hit upon a major insight that we ourselves had yet to realize: in using the difference target propagation formalism and related proof of Lee et al. (2015), we essentially assumed that the forward and backward functions in the network were becoming, roughly, inverses of each other (i.e. that the “… matrix product (J_β) (J_γ) is close to the identity mapping in the readout space…”). Yet, in using random, fixed feedback weights without an inverse loss function to train the feedback, we had no guarantee that this condition actually held.

As reviewer #3 surmised, the answer to this problem lies in the behaviour of the feedforward weights from the hidden layer to the output layer, **W**^1^. As in Lillicrap et al. (2016), we find that **W**^1^ “aligns” with the feedback matrix **Y**. More precisely, we find that as learning proceeds in the first epoch, the maximum eigenvalue of the matrix product (*I* – *J_f_J_g_)(I* – *J_f_J_g_*) drops below 1, thereby meeting the conditions of the Lee et al. (2015) proof for difference target propagation (see Figure 4—figure supplement 1 which contains this new data). (Note, although this is a very important piece of data in our opinion, we put this new figure in the Supplemental Information in consideration of the general audience at eLife – expert readers like this reviewer will want to see it, but most readers will likely find its specific meaning confusing).

We think that this result is exciting, because it shows that feedback alignment from Lillicrap et al. (2016) and difference target propagation from Lee et al. (2015) are intimately linked. As the reviewer suggests, once this theoretical connection is made clear the idea of using segregated dendrites to implement these sorts of deep learning algorithms can be stressed. We now have the following section in the manuscript:

“Interestingly, the correlations between L^0^ and L^1^ were smaller on the first epoch of training. […] Altogether, our model demonstrates that credit assignment using random feedback weights is a general principle that can be implemented using segregated dendrites.”

A bit less fundamental, but still more than minor:2) In view of the rather deep mathematical issues related to the feedback alignment, I would suggest to defer Lemma 1 to some Supplementary Information. The approximation of PSP signaling by instantaneous Poisson rates when the rate is small as compared to the PSP duration is standard in theoretical neuroscience. But the 3-page proof is still nicely done and may be helpful for a non-specialist who wishes to go into the details.

We agree with the reviewer's assessment that this Lemma is useful, but not particularly novel for many theoretical neuroscientists. We have moved it to the back of the Supplemental Information, as recommended.

3) At the end of the subsection “A network architecture with segregated dendritic compartments” (Results) some critical issues are raised about the biological plausibility. In this context it should also be stressed that the alternation between two phases, each of which again subdivided into two further phases (Figure 1), is not so easy to match to the biology. The phases need a memory that is tagged with the phase information and plasticity that is only turned on in a specific phase, checking out the memory from a previous phase.Beside mentioning this in the Results, it should also be taken up in a further paragraph in the Discussion. One should mention that synaptic eligibility traces could help out here and that this helps to bridge information across the phases. Moreover, the phases could be implemented by exploiting global (I guess γ) oscillations that are shown to be present in various cognitive states. Discussing the link of learning and γ oscillations may be of general interest in this context.

Indeed, reviewer #3 is correct that our model requires two different phases (possibly mediated by oscillations) and some form of spatially local temporal storage of information (possibly mediated by synaptic eligibility traces). To make these issues clear for the reader, we have now included the following new sections in the manuscript:

“…it is entirely plausible that neocortical micro-circuits would generate synchronized pyramidal plateaus at punctuated periods of time in response to dis-inhibition of the apical dendrites governed by neuromodulatory signals that determine “phases” of processing. Alternatively, oscillations in population activity could provide a mechanism for promoting alternating phases of processing and synaptic plasticity (Buzsáki and Draguhn, 2004).”

“It should be recognized, though, that although our learning algorithm achieved deep learning with spatially local update rules, we had to assume some temporal non-locality. […] Hence, our model exhibited deep learning using only local information contained within the cells.”

[Editors’ note: the author responses to the re-review follow.]

This paper is much improved. However, it still has a way to go before it's ready for a neuroscience audience. Given that this has been reviewed several times now and remains in an unacceptable form, we are prepared to offer only one more opportunity to provide an acceptable version of the manuscript.The easy thing to fix is notation and writing: we believe that, even in its improved form, it would be very hard for a neuroscientist, even a computational one who is used to thinking about circuits, to read, and the main ideas would be difficult to extract. More on that below.The potentially harder thing to fix is biological plausibility. If we understand things correctly, the neuron must estimate the average PSPs during the feedforward sweep of activity, when only the input is active, estimate them again during the training phase, when the correct output is active as well, and then subtract the two. These signals are separated in time, which means the synapses have to store one signal, wait a few tens of ms, store another, and take the difference. In addition, the difference is computed in the apical dendrite, but it must be transferred to the proximal dendrites. And finally, a global signal is required to tell a synapse in which phase it is so that the estimate can be endowed with the correct sign. All seem nontrivial for neurons and synapses to compute.Lack of biological plausibility should not rule out a theory – synapses and neurons are, after all, complicated. However, two things are needed. First, you need to provide a mechanism for implementing the learning rule that's not inconsistent with what's known about neurons and synapses. Second, you need to provide suggest experiments to test these predictions. Of these, the first is probably harder.Now for the exposition. It may seem like we're micromanaging (and we are), but if this paper is to have an impact on the neuroscience community – a prerequisite for publication in eLife – it has to be cast into familiar notation.1) The model was much simpler, and more standard, than first impressions would imply. It can be written in the very familiar formdV^m_i/dt = -g_L V^m_i+ sum_n g_n (b^n_i + sum_j W^mn_ij s^n_j(t) – V^m_i)+ g^m_iE(t) (E_E – V_i^m) + g^m_iI(t) (E_I – V_i^m)where the s^n_j(t) are filtered spike trains,s^n_j(t) = sum_k kappa(t-t^n_jk),t^n_jk is the k^th spike on neuron j of type n, and spikes were generated via a Poisson process based on the voltage. (Please note: errors are possible, but the equations look something like what we wrote.)In this form it is immediately clear to a neuroscientist what kind of network this is. If nothing else, that will save a huge amount of time for the reader – it took hours of going back and forth over the equations in the paper before it became clear that the model was very standard, something that most readers would not have the patience for.In addition, as written above, it makes it clear exactly how the dendrites are implemented: by varying the g_n. In real dendrites they vary with voltage on the dendrite; in this model they simply vary with time.And finally, the notation with the A's and B's used in the paper is not helpful to neuroscientists, who are very used to seeing V, or maybe U, for voltage.2) Along the same lines, a better figure showing the circuit needs to be included. The circuit with multiple hidden layers needs a similar drawing, as we were not able to figure out exactly what it looked like. (We're guessing there was sufficient information in the paper, but the amount of work it would take to extract it seemed high.)3) The cost function, L^1, seemed somewhat arbitrary. According to Equation 7,L^1 \propto \sum_i (<σ(U_i)>^t – σ(<U_i>^f))^2where the angle brackets represent a time average and the superscripts t and f refer to the target and feedforward phases, respectively (basically, the overline was replaced with angle brackets, mainly because we're using plain text). Why was the average taken outside the sigmoid in the target phase and inside the sigmoid in the feedforward phase?4) A similar question applies to L^0, which is writtenL^0 = sum_i (λ_max(<σ(C_i)>^f – σ(<C_i>^f) + α_i^t – α_i^f)^2As far as we can tell, the first two terms are included to make the update rules work out, and they are eventually set equal to each other. But is there any reason to think that L^0 should be minimized? It seemed unmotivated.5) In Equations 19 and 22, why are there no terms involving the derivatives of the sigmoid in the target phase?

In our last submission, we only received feedback from one reviewer. That reviewer was still concerned about the ease with which the paper could be understood by a general neuroscience audience. With the help of one of the editors, we have worked hard to make the paper easier to understand. We believe that the paper has improved immensely as a result. Now, we explain the dynamics of our simulations in a clear manner that would make it easy to replicate them, and we provide a far more intuitive explanation of how we solve the credit assignment problem with our loss functions. We have also redone the figures, and added a final figure illustrating the model’s experimental predictions. As a result, we believe that the paper is now appropriate for a general neuroscience audience, and we hope you agree.